# COUNTERFACTUAL GENERATIVE MODELING WITH VARIATIONAL CAUSAL INFERENCE

**Yulun Wu**
University of California, Berkeley
yulun_wu@berkeley.edu

**Louie McConnell**
Genentech
mcconnl3@gene.com

**Claudia Iriondo**
Genentech
iriondoc@gene.com

## ABSTRACT

Estimating an individual's counterfactual outcomes under interventions is a challenging task for traditional causal inference and supervised learning approaches when the outcome is high-dimensional (e.g. gene expressions, facial images) and covariates are relatively limited. In this case, to predict one's outcomes under counterfactual treatments, it is crucial to leverage individual information contained in the observed outcome in addition to the covariates. Prior works using variational inference in counterfactual generative modeling have been focusing on neural adaptations and model variants within the conditional variational autoencoder formulation, which we argue is fundamentally ill-suited to the notion of counterfactual in causal inference. In this work, we present a novel variational Bayesian causal inference framework and its theoretical backings to properly handle counterfactual generative modeling tasks, through which we are able to conduct counterfactual supervision end-to-end during training without any counterfactual samples, and encourage disentangled exogenous noise abduction that aids the correct identification of causal effect in counterfactual generations. In experiments, we demonstrate the advantage of our framework compared to state-of-the-art models in counterfactual generative modeling on multiple benchmarks.

## 1 INTRODUCTION

In traditional causal inference, heterogeneous treatment effect is typically formulated as the interventional model $p(Y|X, \mathrm{do}(T))$ (or $\mathbb{E}[Y|X, \mathrm{do}(T)]$) with outcome $Y$, covariates $X$, treatment $T$, and estimated by the observed covariate-specific efficacy $p(Y|X, T)$ under the ignorability assumption (Rosenbaum & Rubin, 1983; Pearl, 1995). However, in cases such as the single-cell perturbation datasets (Dixit et al., 2016; Norman et al., 2019; Schmidt et al., 2022) where $Y$ has thousands of dimensions while $X$ has only two or three categorical features, such model could hardly be relied on to produce useful individualized results. To construct outcome $Y'$ under a counterfactual treatment $T'$ in such high-dimensional outcome scenario, it is important and necessary to learn the individual-specific efficacy $p(Y'|Y, X, T, T')$ conditioning on the factual outcome $Y$ itself, which leverages the rich information embedded in the factual outcome that cannot be recovered by the handful of covariates. For example, given a cell with type A549 ($X$) that received SAHA drug treatment ($T$), we may want to know what its gene expression profile ($Y'$) would have looked like if it had received Dex drug treatment ($T'$) instead. In this case, we would want to take the profiles of other cells with type A549 that indeed received Dex as reference, such that the counterfactual construction $Y'$ exhibits the treatment characteristics of Dex on A549, but would also want to extract as much individual information as possible from its own expression profile ($Y$) such that the counterfactual construction could preserve this cell's individuality such as cell state.

Yet the lack of observability of counterfactual outcome $Y'$ makes this objective intractable and hence presents a major difficulty for supervised learning when $Y$ is taken as an input. In previous works that involve self-supervised counterfactual generation (Kocaoglu et al., 2017; Louizos et al., 2017; Yoon et al., 2018; Pawlowski et al., 2020; Yang et al., 2021; Kim et al., 2021; Lotfollahi et al., 2021; Sauer & Geiger, 2021; Feng et al., 2022; Shen et al., 2022) based on Variational Autoencoders (VAE) (Kingma & Welling, 2013) and Generative Adversarial Networks (GAN) (Goodfellow et al., 2014), there is no counterfactual supervision during training, and hence no explicit regulation on the trade-off between treatment characteristics and individuality. When such regulation does not exist, there is

a disconnection between training and counterfactual inference. To see this, consider a conditional generative model $g_\theta(y, t)$ that is only supervised on the reconstruction loss with respect to outcome $y$ during training. In this case, the model is free to ignore the extra treatment $t$ and converge to a state such that $g_\theta(y, t) = g_\theta(y, 0)$ (Chen et al., 2016), hence preserving maximum individuality yet minimum treatment characteristics in counterfactual construction $g_\theta(y, t')$ during inference. This is particularly an issue for high-fidelity counterfactual generation with state-of-the-art Hierarchical Variational Autoencoders (HVAE) (Vahdat & Kautz, 2020; Child, 2020), as the bottleneck latent representations possess much larger number of dimensions compared to the observed outcome $y$. In more recent works on counterfactual generative modeling with HVAEs and diffusion models (Sanchez & Tsaftaris, 2022; Monteiro et al., 2023; Ribeiro et al., 2023), only Ribeiro et al. (2023) touched on the issue of counterfactual supervision, yet its proposed method does not have a theoretical backing in variational inference, nor could it be conducted end-to-end during training. As we will see in the proceeding sections of this work, there is a fundamental incompatibility of the conditional VAE formulation with counterfactual generative modeling, which is the root cause of such issue. In fact, this incompatibility also exhibits itself in another important aspect of counterfactual generative modeling which is latent disentanglement – the partially abducted exogenous noise (i.e. shallow-level latent representation) in conditional HVAEs and diffusion models is able to encode individuality better than traditional VAEs, but at the same time is still heavily entangled with treatment variables and often result in lingering characteristics of the factual treatment in counterfactual construction (Monteiro et al., 2023; Sanchez & Tsaftaris, 2022). More introduction regarding latent disentanglement can be found in Appendix A.

Our contributions in this work are summarized as follow: **1)** we propose a formulation and stochastic optimization scheme for counterfactual generative modeling by specifically formulating counterfactual variables and using variational inference to derive the evidence lower bound (ELBO) for the individual-level likelihood $p(Y'|Y, X, T, T')$ – a well-motivated and fitting objective for counterfactual generative modeling instead of the marginal-level (interventional) conditional likelihood $p(Y|X, T)$ or even joint likelihood $p(Y, X, T)$ used by VAE-based prior works. **To emphasize, the contribution here is not to propose another model variant under the conditional VAE formulation such as conditional HVAEs or diffusion models (Sanchez & Tsaftaris, 2022; Monteiro et al., 2023; Ribeiro et al., 2023), but to fundamentally change the VAE formulation.** We call this framework Variational Causal Inference (VCI). An explanation to why the conditional VAE formulation (including diffusion models) is ill-suited to counterfactual generative modeling and a straightforward comparison between VAE's formulation and ours can be found in Appendix C. **2)** Our proposed optimization scheme allows us to conduct **counterfactual supervision end-to-end during training** alongside self-supervision, even without any counterfactual observations, which has not been done in prior works to the best of our knowledge. **3)** This workflow of constructing and supervising counterfactual outcomes during training presents us the unique opportunity to naturally enforce **disentangled exogenous noise abduction** through distribution alignment on matching pairs (Shu et al., 2019; Locatello et al., 2020; Brehmer et al., 2022) without having paired data, which has not been done in counterfactual generative modeling, to the best of our knowledge. **4)** We further propose a robust estimation scheme for high-dimensional marginal causal parameters leveraging this individual-level counterfactual construction framework. No prior work in counterfactual generative modeling conducts such asymptotically efficient marginal estimation upon acquiring individual predictions, to the best of our knowledge. **5)** In experiments, our proposed method is evaluated on datasets with vector outcomes – single cell perturbation datasets, as well as datasets with image outcomes – facial imaging and handwritten digits datasets, and compared to state-of-the-art models in the two domains. The results show that ours outperformed state-of-the-arts in both domains with notable margins. Related work and comparative analysis can be found in Appendix B.

## 2 PROPOSED METHOD

### 2.1 FORMULATION AND INTUITION

Let $Y : \Omega \to (\mathcal{Y}, \Sigma_\mathcal{Y})$ be the outcome, $X : \Omega \to (\mathcal{X}, \Sigma_\mathcal{X})$ be the covariates, $T : \Omega \to (\mathcal{T}, \Sigma_\mathcal{T})$ be the treatments and $Z : \Omega \to (\mathcal{Z}, \Sigma_\mathcal{Z})$ be a latent feature vector on probability space $(\Omega, \Sigma, P)$. Suppose the causal relations between random variables (and random vectors) follow a structural causal model (SCM) (Pearl, 1995) defined in Figure 1, where we adopt twin networks (Balke & Pearl, 1994) to formulate counterfactuals $Y'$ and $T'$ as separate variables apart from $Y$ and $T$ having

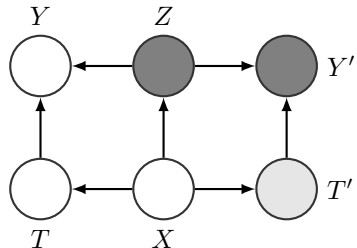 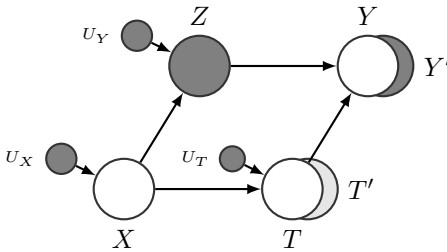

(a) Bayesian network      (b) Counterfactual view of the Bayesian network

Figure 1: The causal diagram where variables are generated from the following SCM: $X = f_X(U_X)$; $T = f_T(X, U_T)$, $T' = f_T(X, U'_T)$ where $U_T$, $U'_T$ are i.i.d.; $Z = f_Z(X, U_Y)$; $Y = f_Y(Z, T, \epsilon_Y)$, $Y' = f_Y(Z, T', \epsilon'_Y)$ where $\epsilon_Y$, $\epsilon'_Y$ are i.i.d., independent of all $U$s, and are constants w.p. 1 if the consistency assumption is assumed. Endogenous variables follow the Bayesian network on the left under the ignorability assumption, with permissibly one additional edge from either $Z$ or $T$ to $X$ if $U_Y$ or $U_T$ is dependent of $U_X$. We call $O = (X, T, Y)$ observed variables, $B = (X, T, T', Y)$ sample variables ($T'$ are sampled for model training and evaluation), $D = (X, T, T', Y, Y')$ full data variables and $W = (Z, X, T, T', Y, Y')$ all variables. White nodes are observed, light grey nodes are assigned during training and inference, dark grey nodes are unobserved.

a conditional distribution $p(Y', T'|Z, X)$ identical to that of its factual counterpart $p(Y, T|Z, X)$. Different from Vlontzos et al. (2023), we introduce latent $Z$ in high-dimensional outcome settings as a feature vector of $Y$ such that random factor $\epsilon_Y$ does not capture any more uncertainty that can be retrieved from its counterfactual counterpart, i.e. $\epsilon_Y \perp\!\!\!\perp Y'$ and similarly $\epsilon'_Y \perp\!\!\!\perp Y$. The consistency and ignorability assumptions corresponding to this causal formulation can be found in Appendix H. Under the consistency assumption (Assumption 2 and Remark 2), $\epsilon_Y = 0$ and there is no more uncertainty in $Y$ beyond $Z$, i.e. the shared random factor $U_Y$ fully captures the exogenous noise injected to $Y$ and $Y'$. Hence, $Z$ in this setting can be seen as an unobserved summary random vector of the covariates $X$ and exogenous noise $U_Y$. Note that the consistency assumption is not generally required for our subsequent variational inference, but is required for $Y'_{do(T')}$ to be formally defined as the counterfactual of $Y$ under the three-layer causal hierarchy of Pearl (2009). The ignorability assumption (Assumption 1 and Remark 1) is generally required, under which $do(T' = t')$ and $T' = t'$ result in the same outcome distribution on $Y'$ conditioned on $X$ or $Z$. See Appendix D for the connection between our formulation and the traditional SCM formulation.

**Semi-autoencoding** Under this formulation, $Z$ has a posterior distribution $p(Z|Y, T, X)$ given the observed variables; $Y$ and $Y'$ can be constructed by $p(Y|Z, T)$ and $p(Y'|Z, T')$ respectively with latent features $Z$. Similar to prior works in autoencoder (Vincent et al., 2008; Bengio et al., 2014), we can estimate the latent recognition model (i.e. exogenous noise abduction model) and outcome construction model with deep neural network encoder $q_\phi$ and decoder $p_\theta$. Given a sample $b = (x, t, t', y)$, while the reconstruction $y_{\theta,\phi}$ of $y$ is self-supervised, we can only assess the counterfactual construction $y'_{\theta,\phi}$ under $t'$ by looking at its resemblance to similar individuals that indeed received treatment $t'$. Hence, naively, we may conduct counterfactual supervision during training with a semi-autoencoding (SAE) loss function as follow

$$L_{\theta,\phi}(b) = L_2(y_{\theta,\phi}, y) - \omega \cdot \ell_{\hat{p}(Y'|x,t')}(y'_{\theta,\phi}) \tag{1}$$

where $\omega$ is a scaling coefficient and $\hat{p}$ is the traditional covariate-specific outcome model fit on the observed variables $(X, T, Y)$ (notice that $p(Y'|x, T' = t') = p(Y|x, T = t')$). The intuition is that, if $y'_{\theta,\phi}$ is indeed one's outcome under $t'$, then the likelihood of $y'_{\theta,\phi}$ coming from the outcome distribution of individuals with the same attributes $x$ that factually received treatment $t'$ should be high. In practice, we can fit $\hat{p}(Y|X, T)$ in an end-to-end fashion using a discriminator $\mathcal{D}(X, T, Y)$ on factual triplets $(x, t, y)$ and counterfactual triplets $(x, t', y'_{\theta,\phi})$ with the adversarial approach (Goodfellow et al., 2014). Specifically, use discriminator loss $L_\mathcal{D}(b, y'_{\theta,\phi}) = -\log[\mathcal{D}(x, t, y)] - \log[1 - \mathcal{D}(x, t', y'_{\theta,\phi})]$ to train model $\mathcal{D}$, and use generator loss $\ell_{\hat{p}(Y'|x,t')}(y'_{\theta,\phi}) = \log \mathcal{D}(x, t', y'_{\theta,\phi})$ in $L_{\theta,\phi}(b)$ (notice that $p(X, T) = p(X, T')$). This end-to-end approach prevents $(p_\theta, q_\phi)$ from exploiting a pre-trained $\hat{p}$ model. In cases where covariates are limited and discrete such as the

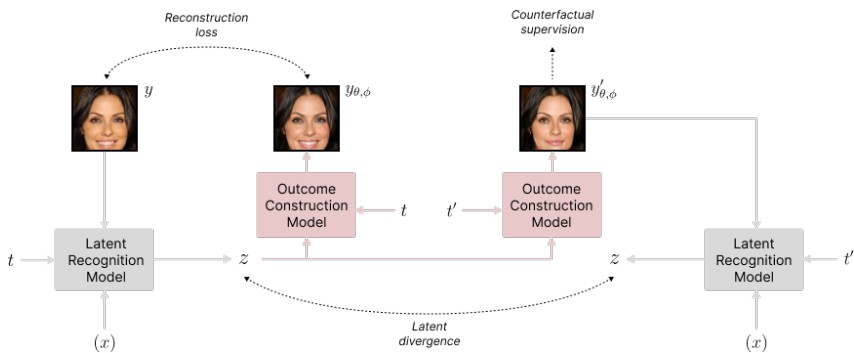

Figure 2: Model workflow of the Variational Causal Inference (VCI) framework. In a forward pass, the encoding model takes outcome $y$ as well as its treatments $t$ and covariates $x$ (if any) as inputs and attains latent feature $z$; $(z, t)$ and $(z, t')$ where $t'$ is the counterfactual treatments are separately passed into the decoding model to attain reconstruction $y_{\theta,\phi}$ and counterfactual construction $y'_{\theta,\phi}$, for which reconstruction loss and counterfactual supervision loss are evaluated; $y'_{\theta,\phi}$ is then passed back into the encoding model along with $t'$ and $x$ to attain $z$ again, which is encouraged to match the latent feature $z$ previously acquired from encoding the factual outcome $y$ and treatment $t$ through the latent divergence term. The neural architectures of the latent recognition and outcome construction models for image generation tasks can be found in Appendix J.

single-cell perturbation datasets, $\hat{p}(Y|X, T)$ can simply be a smoothed empirical outcome distribution under treatment $T$ stratified by covariates $X$. The SAE approach motivates the derivation of our main theorem presented in the next section, and serves as a baseline comparison to our main algorithm in the ablation study of the experiments section.

## 2.2 VARIATIONAL CAUSAL INFERENCE

Now we present our main result by rigorously formulating the objective, providing a probabilistic theoretical backing, and specifying an optimization scheme that reflects the intuition described in the previous section. Suppose we want to optimize $p(Y'|Y, X, T, T')$ instead of the traditional outcome likelihood $p(Y|X, T)$. The following theorem induces the evidence lower bound (ELBO) and thus provides a roadmap for stochastic optimization:

**Theorem 1.** *Suppose a collection of random variables $W$ follows a causal structure defined by the Bayesian network in Figure 1. Then $J(D) = \log[p(Y'|Y, X, T, T')]$ has the following variational lower bound:*

$$
\begin{aligned}
J(D) \geq \mathbb{E}_{p(Z|Y,T,X)} \log \left[p(Y|Z, T)\right] - D\left[p(Y|X, T) \parallel p(Y'|X, T')\right] \\
- D_{\mathrm{KL}}\left[p(Z|Y, T, X) \parallel p(Z|Y', T', X)\right]
\end{aligned}
\tag{2}
$$

*where $D[p \parallel q] = \log p - \log q$.*

Proof of Theorem 1 and proofs of all theoretical results in the following sections can be found in Appendix I. Theorem 1 suggests that, in order to maximize the counterfactual outcome likelihood $p(Y'|Y, X, T, T')$, we can maximize the following estimated ELBO parameterized with estimating models $p_\theta$ and $q_\phi$ given each sample $b = (x, t, t', y)$:

$$
\begin{aligned}
J_{\theta,\phi}(b) = \mathbb{E}_{q_\phi(z|y,t,x)} \log \left[p_\theta(y|z, t)\right] + \log \left[\hat{p}(y'_{\theta,\phi}|x, t')\right] \\
- D_{\mathrm{KL}}\left[q_\phi(z|y, t, x) \parallel q_\phi(z|y'_{\theta,\phi}, t', x)\right],
\end{aligned}
\tag{3}
$$

where $y'_{\theta,\phi} = \mathbb{E}_{p_\theta(\cdot|z_\phi,t')}Y' = \mathbb{E}_{p_\theta(\cdot|\mathbb{E}_{q_\phi(\cdot|y,t,x)}Z,t')}Y'$ and $\hat{p}$ can be estimated with the approaches described at the end of Section 2.1. Note that $\hat{p}(y|x, t)$ does not impose gradient on $(p_\theta, q_\phi)$ and is thus omitted in $J_{\theta,\phi}(b)$. The stochastic optimization of ELBO is hence conducted on the objective $J_{p_{\mathrm{data}}}(p_\theta, q_\phi) = \mathbb{E}_{p_{\mathrm{data}}} J_{\theta,\phi}(b)$ where $t' \sim p_{\mathrm{data}}(T|x)$. A figure demonstrating the workflow of this stochastic optimization scheme can be found in Figure 2.

As can be seen from Equation 3, the ELBO consists of the individual-specific factual outcome likelihood $\mathbb{E}_{q_\phi(z|y,t,x)} \log[p_\theta(y|z, t)]$ and the covariate-specific counterfactual outcome likelihood

$\log[p(y'_{\theta,\phi}|x, t')]$, echoing the intuition highlighted by Equation 1, with an additional divergence term $-D_{\mathrm{KL}}[q_\phi(z|y, t, x) \parallel q_\phi(z|y'_{\theta,\phi}, t', x)]$ that regularizes the similarity across latent distributions under different potential scenarios. Note that this framework is fundamentally different from VAE formulations such as CVAE (Sohn et al., 2015) or CEVAE (Louizos et al., 2017), since the variational lower bound is derived directly from the causal objective and the KL-divergence term serves causal purposes that are entirely different from the prior latent bounding purpose of VAEs. An interpretation of this divergence term is given in the paragraph below. A comparison between VCI and conditional VAE using standard VAE notations is given in Appendix C. A summary of different optional gradient detaching patterns for training stability is given in Appendix K. An approach to conduct counterfactual supervision implicitly using traditional variational inference is given in Appendix E.

**Divergence Interpretation**   The divergence term $-D_{\mathrm{KL}}[q_\phi(z|y, t, x) \parallel q_\phi(z|y'_{\theta,\phi}, t', x)]$ is the key of the framework and it serves two purposes. Firstly, it adds robustness to latent encoding by encouraging the encoding model to recognize individual features contained in both factual and counterfactual outcomes. In fact, these common features are well disentangled from the treatment variables, which we will discuss in-depth in the next section. Secondly, it encourages the preservation of individuality in counterfactual outcome constructions. To see this, notice that if the counterfactual outcome construction was only penalized by the covariate-specific likelihood loss $-\log[\hat{p}(y'_{\theta,\phi}|x, t')]$, the counterfactual decoding model $p_\theta(Y'|z_\phi, t')$ could learn to completely discard the identity of individual subjects represented in latent features $z_\phi$ once it detects a counterfactual treatment $t' \neq t$ in the inputs. Such behavior is regulated by the divergence term, since otherwise we would not be able to recover a latent distribution $q_\phi(Z|y'_{\theta,\phi}, t', x)$ close to $q_\phi(Z|y, t, x)$.

**Robust Marginal Estimation**   Similar to the augmented inverse propensity weighted (AIPW) estimator in traditional causal inference, the individual-level model predictions acquired by our framework can also aid the robust estimation of average treatment effect $\Psi(p) = \mathbb{E}_p[Y'_{\mathrm{do}(T'=\alpha)}]$ for a treatment level $\alpha$ of interest if such estimation is desired. Under the formulation in Figure 1, we have the following robust estimator that is asymptotically efficient under some regularity conditions (Van Der Laan & Rubin, 2006):

$$\hat{\Psi}_n(o) = \frac{1}{n} \sum_{k=1}^{n} \left\{ \frac{I(t_k = \alpha)}{\hat{e}(t_k|x_k)} \cdot y_k + \left(1 - \frac{I(t_k = \alpha)}{\hat{e}(t_k|x_k)}\right) \cdot \mathbb{E}_{p_\theta}\left[Y'|z_{k,\phi}, T' = \alpha\right] \right\}, \qquad (4)$$

where $o_k = (x_k, t_k, y_k) \overset{\mathrm{iid}}{\sim} p(x, t, y)$ is the vector of observed triplet of the $k$-th individual, $z_{k,\phi} \sim q_\phi(y_k, t_k, x_k)$, and $\hat{e}$ is an estimation of the propensity score that satisfies the positivity assumption (Robins, 1986). See Appendix G for the derivation of this estimator. Notice that the regression adjustment term $\mathbb{E}_{p_\theta}[Y'|z_{k,\phi}, T' = \alpha]$ in ours has the ability to vary across different individuals within the same covariate group, which is not the case for that of the AIPW estimator in traditional causal formulation. Hence, it is meaningful to conduct the robust estimation of covariate-specific marginal treatment effect $\Xi(p) = \mathbb{E}_p[Y'_{\mathrm{do}(X=c, T'=\alpha)}]$ for a given covariate $c$ of interest, see Appendix G for details.

## 2.3   Latent Identifiability and Exogenous Noise Disentanglement

In any deep probabilistic model, there is a certain degree of freedom and arbitrarity to the learnt latent distribution, even with latent restrictions. Naturally, one might question the latent identification of our variational causal framework: when identifying counterfactual $Y'$ under $T'$, does the learnt latent $Z$ inevitably carry over some characteristics of the factual treatment in $Y$ and $T$? Ideally, we would like to aid counterfactual construction by attaining a clean disentanglement between the learnt latent and the observed treatment, such that the counterfactual generation does not preserve influence from the factual treatment. Such disentangled exogenous noise abduction fits the semantic of our generating process (Figure 1), where $Z$ and $T$ are not descendant of each other and $Z \perp\!\!\!\perp T|X$, as well as the semantic of causal identification in the traditional formulation, where the identifiability of $Y_{\mathrm{do}(T)}$ relies on the ignorability assumption that exogenous noise does not contain unobserved confounders. In this section, we show that optimizing the VCI framework in fact grants such disentanglement between $Z$ and $T$ under mild assumptions, leading to a more in-depth understanding of the purpose and behavior of the latent divergence term in VCI's optimization scheme.

Similar to Shu et al. (2019), we formally define and measure disentanglement in terms of oracle consistency and oracle restrictiveness, except that our definitions are generalized to the case of stochastic models and models with auxiliary attributes. For notation simplicity of the oracle operation on functions with auxiliary inputs, we use $\gamma \circ \zeta(a; b)$ to denote $\gamma(c, b)$ where $c = \zeta(a)$ if $\zeta$ is a deterministic function and $c \sim \zeta(a)$ if $\zeta$ is a probability measure. Denote the true data generating distribution as $p_0$. The definitions of oracle consistency, oracle restrictiveness, and disentanglement are then given in Appendix F.

Consider our model class $\{(p_., q_.) \mid p_. : \mathcal{Z} \times \mathcal{T} \to [0,1]^{\Sigma_y}, \ q_. : \mathcal{Y} \times \mathcal{T} \times \mathcal{X} \to [0,1]^{\Sigma_z}\}$ and let $S := (Z, T)$, $V := (Y, T)$. We regard $T$ as part of the model outputs such that the oracle operations above are well-defined, i.e. let $\mathcal{H} = \{(\tilde{p}_., \tilde{q}_.) \mid \tilde{p}_.(s) = p_.(Y|z, t) \times \delta_t(T), \ \tilde{q}_.(v, x) = q_.(Z|y, t, x) \times \delta_t(T)\}$ where $\delta_t$ is the Dirac measure ($\delta_t(T) = 1$ if $T = t$ else 0). Then we have the following lemma:

**Lemma 1.** *If $q$ minimizes the latent divergence in Equation 2 over $p_0(D)$, i.e.*

$$\mathbb{E}_{p_0} D_{\mathrm{KL}} \left[ q(z|y, t, x) \parallel q(z|y', t', x) \right] = 0, \tag{5}$$

*then $\tilde{q}$ disentangles $Z$ and $T$.*

Lemma 1 builds a direct connection between the latent divergence term and disentangled abduction of exogenous noise. Intuitively, it states that if the encoder encodes the same latent representation for any pair of potential outcomes of the same individual, then the latent must have been disentangled from the treatment. However, does optimizing the ELBO necessarily imply that the latent divergence term reaches minimum? We proceed to show that this can be implied under mild assumptions.

**Proposition 1.** *With the true distribution $p_0$ satisfying Assumption 2 in Appendix H and Assumption 3 in Appendix I.2.2, if $(p_*, q_*)$ maximizes the evidence lower bound over $p_0(D)$ and any $\hat{p}$, i.e.*

$$(p_*, q_*) \in \underset{p_., q_.}{\arg\max} \mathbb{E}_{p_0} \left\{ \mathbb{E}_{q_.(z|y, t, x)} \log \left[ p_.(y|z, t) \right] + \log \left[ \hat{p}(y'|x, t') \right] \right.$$
$$\left. - D_{\mathrm{KL}} \left[ q_.(z|y, t, x) \parallel q_.(z|y', t', x) \right] \right\} \tag{6}$$

*then $\tilde{q}_*$ disentangles $Z$ and $T$.*

Intuitively, Proposition 1 states that as long as the treatments do not have any unknown side effect and do not completely wipe out different features in different individuals, the solution to maximizing the ELBO disentangles $Z$ and $T$. In practice, we can formulate model class as deterministic functions (i.e. degenerate distributions) plus noise similar to prior work, and encoder noise as standard normal:

$$\{p_. \mid g_.(Y|z, t) := \delta_{\mathbb{E}_{p_.(\cdot|z, t)} Y}, \ \epsilon_.(z, t) := p_.(Y - \mathbb{E}_{p_.(\cdot|z, t)} Y|z, t)\} \tag{7}$$

$$\{q_. \mid e_.(Z|y, t, x) := \delta_{\mathbb{E}_{q_.(\cdot|y, t, x)} Z}, \ q_.(Z - \mathbb{E}_{q_.(\cdot|y, t, x)} Z|y, t, x) = \mathcal{N}(0, 1)\} \tag{8}$$

and learn model $e_.$, $g_.$ and $\epsilon_.$ during training. The following proposition states that disentanglement under the estimating model's own oracle can be achieved under this setting by optimizing the VCI objective in real-world optimizations, where neither the true generating mechanisms nor the true factual-counterfactual pairs are available.

**Proposition 2.** *Given dataset $\{(x_k, t_k, y_k)\}|_{k=1}^n$, let $p_{\mathrm{data}}$ be its empirical distribution and the model class in Equation 7 and Equation 8 satisfy the following uniqueness conditions:*

*1) $\forall \alpha \in \mathcal{T}$, we have $g_.(Y|z, \alpha) \neq g_.(Y|z', \alpha)$ if $z \neq z'$;*

*2) $\forall x, t, y, t' : p_{\mathrm{data}}(x, t, y) \cdot p_{\mathrm{data}}(t'|x) > 0$ and $y' = \mathbb{E}_{g_.(\cdot|z, t')e_.(z|y, t, x)} Y$, we have $p_{\mathrm{data}}(x, t', y') = 0$ if $(x, t, y) \neq (x, t', y')$.*

*If $(p_*, q_*)$ maximizes the VCI objective in Equation 3 over $p_{\mathrm{data}}$ and any $\hat{p}$, i.e.*

$$(p_*, q_*) \in \underset{p_., q_.}{\arg\max} J_{p_{\mathrm{data}}}(p_., q_.) \tag{9}$$

*then $(\tilde{g}_*, \tilde{q}_*)$ disentangles $Z$ and $T$ under $\int_y e_* p_{\mathrm{data}}$.*

We note that uniqueness condition 2) in Proposition 2 is reasonable in high-dimensional outcome settings: taking the facial imaging dataset as an example, it wouldn't make sense for a model's constructed image for a certain individual to be an exact match of the factual outcome of another individual in the dataset, as long as the dataset does not contain multiple images of the same individual in the same environment.

**Corollary 1.** *Under the same settings and conditions of Proposition 2, $(\tilde{g}_*, \tilde{e}_*)$ disentangles $Z$ and $T$ under $\int_y e_* p_{\text{data}}$.*

Corollary 1 states that the optimal model is guaranteed to disentangle $Z$ and $T$ under the constructed latent distribution in inference mode without noise injection. In summary, the disentanglement results in our framework rely on the simple fact that the latent divergence term is established on an individual level, and does not necessarily compete with outcome supervisions under mild assumptions. Prior works in VAEs have proposed bounding latent on conditional marginal prior (Sohn et al., 2015; Khemakhem et al., 2020), and VCI can be seen as a further relaxed form of latent restriction in the sense that the latent for any given subject need not be restricted to any prior distribution – we only require the latent distributions of a subject's different potential outcomes to match each other. More insights on how the latent divergence term helps the training of VCI in practice can be found in the ablation study in the experiments section.

## 3 EXPERIMENTS

We present experiment results of our framework on two datasets with vector outcomes (sci-Plex dataset from Srivatsan et al. (2020) (Sciplex) and the CRISPRa dataset from Schmidt et al. (2022) (Marson)) and two datasets with image outcomes: Morpho-MNIST (Castro et al., 2019) and CelebA-HQ (Karras et al., 2017). Results from the former are compared to state-of-the-art models in single-cell perturbation prediction and results from the latter are compared to state-of-the-art models in counterfactual image generation. In both cases, our model exhibited superior performance against benchmarks. Details of model and dataset settings for each experiment can be found in Appendix M.

### 3.1 SINGLE-CELL PERTURBATION DATASETS

Same as Lotfollahi et al. (2021), we evaluate our model and benchmarks on a widely accepted and biologically meaningful metric — the $R^2$ (coefficient of determination) of the average prediction against the true average from the out-of-distribution (OOD) set on all genes and differentially-expressed (DE) genes. Definitions of OOD set and DE genes can be found in Appendix M.1. Results over five independent runs are shown in Table 1.

Table 1: $\bar{R}^2$ of OOD predictions on single-cell perturbation datasets. AE is a naive baseline adapting Autoencoder to counterfactual generation, see Appendix L.1. GANITE is GANITE's counterfactual block adapting to high-dimensional outcome plus multi-level treatment (see Appendix L.2).

| | Sciplex | | Marson | |
|---|---|---|---|---|
| | all genes | DE genes | all genes | DE genes |
| AE | $0.740 \pm 0.043$ | $0.421 \pm 0.021$ | $0.804 \pm 0.020$ | $0.448 \pm 0.009$ |
| CEVAE (Louizos et al., 2017) | $0.760 \pm 0.019$ | $0.436 \pm 0.014$ | $0.795 \pm 0.014$ | $0.424 \pm 0.015$ |
| GANITE (Yoon et al., 2018) | $0.751 \pm 0.013$ | $0.417 \pm 0.014$ | $0.795 \pm 0.017$ | $0.443 \pm 0.025$ |
| CPA (Lotfollahi et al., 2021) | $\mathbf{0.836} \pm 0.002$ | $0.474 \pm 0.014$ | $0.876 \pm 0.005$ | $0.549 \pm 0.019$ |
| VCI | $0.832 \pm 0.008$ | $\mathbf{0.496} \pm 0.011$ | $\mathbf{0.891} \pm 0.007$ | $\mathbf{0.658} \pm 0.040$ |

For each perturbation of each covariate level (e.g. each cell type of each donor) in the OOD set, the $R^2$ (coefficient of determination) is computed with the average outcome predictions for all genes and DE genes using samples from the validation set against the true empirical average over samples from the OOD set. The average $R^2$ over all perturbations of all covariate levels is then calculated as the evaluation metric $\bar{R}^2$. In these experiments, our variational Bayesian causal inference framework excelled state-of-the-art models in both experiments, with the largest fractional improvement on DE genes which are most causally affected by the perturbations. **Results of marginal estimation using the robust estimator can be found in Appendix N.1.**

### 3.2 MORPHO-MNIST

In an ideal world, we would like to evaluate model performance against benchmarks simply by measuring the error between the counterfactual prediction and the counterfactual truth for each

Table 2: The errors between counterfactual predictions and counterfactual truths on Morpho-MNIST. Images are scaled to $[0, 1]$ in evaluations. On top of randomly sampling counterfactual treatment, we also randomly sample the ratio of modification, which could be any value such that the thickness and intensity of the counterfactual truth is in range. **The standard error of runs and the violin plot of errors are reported in Appendix N.5.**

| | $\beta$ | Image MSE $\downarrow$ ($\cdot 10^{-2}$) | | | Thickness (th) MAE $\downarrow$ | | | Intensity (in) MAE $\downarrow$ ($\cdot 10^{-1}$) | | |
| --- | --- | --- | --- | --- | --- | --- | --- | --- | --- | --- |
| | | $do$(th) | $do$(in) | mix | $do$(th) | $do$(in) | mix | $do$(th) | $do$(in) | mix |
| DEAR (Shen et al., 2022) | | 6.93 | 3.21 | 5.08 | 0.82 | 0.74 | 0.85 | 6.23 | 3.39 | 4.61 |
| Diff-SCM (Sanchez & Tsaftaris, 2022) | | 7.72 | 4.90 | 6.30 | 0.70 | 0.76 | 0.73 | 3.48 | 2.17 | 2.80 |
| CHVAE (Monteiro et al., 2023) | 1 | 4.92 | 4.81 | 4.81 | 0.61 | 0.66 | 0.62 | 3.23 | 1.88 | 2.81 |
| CHVAE (Monteiro et al., 2023) | 3 | 6.93 | 6.19 | 6.40 | 0.71 | 1.34 | 0.97 | 2.37 | 1.31 | 1.89 |
| MED (Ribeiro et al., 2023) | 1 | 2.76 | 0.54 | 1.65 | 0.54 | 0.35 | 0.46 | 1.17 | 0.43 | 0.78 |
| MED (Ribeiro et al., 2023) | 3 | 2.20 | 0.61 | 1.39 | 0.31 | 0.29 | 0.33 | 0.44 | 0.34 | 0.40 |
| SAE (Section 2.1) | | 0.67 | 0.20 | 0.47 | 0.32 | 0.21 | 0.29 | 0.52 | 0.26 | 0.41 |
| VCI (Section 2.2) | | **0.42** | **0.13** | **0.36** | **0.20** | **0.14** | **0.22** | **0.42** | **0.22** | **0.33** |

individual. When the counterfactual truth is attainable, there is no need to resort to approximation metrics such as Monteiro et al. (2023); Melistas et al. (2024). For this reason, we specifically chose the Morpho-MNIST (Castro et al., 2019) dataset to present our main evaluation results because the counterfactual truth can be directly computed on this dataset based on the intervention (modifying thickness or modifying intensity of hand-written digits) even with the existence of exogenous noise. Similar to Ribeiro et al. (2023), we evaluate counterfactual constructions under single modifications as well as mix of modifications, and compare them to state-of-the-art models on high-fidelity image counterfactual generation. Contrary to Ribeiro et al. (2023), the magnitude of modification is randomly sampled, which makes the task significantly harder. The results are shown in Table 2 (standard error in Table 8) which demonstrate that ours beat state-of-the-arts by a wide margin. To understand why conditional diffusion model such as Diff-SCM (Sanchez & Tsaftaris, 2022) does not perform well in counterfactual generative modeling, see Appendix C.1. Note that Ribeiro et al. (2023) only evaluates mean absolute error (MAE) on the thickness and intensity of counterfactual constructions, which took treatment characteristics into account but completely left out how much individuality of the original images has been preserved, whereas mean squared error (MSE) on the image is a comprehensive metric that takes both factors into account and should be the primary evaluation metric when the counterfactual truth is available. **A model inspection using axiomatic soundness metrics (Monteiro et al., 2023) can be found in Appendix N.4.**

To investigate the impact of counterfactual supervision and latent divergence terms in optimizations, we present an ablation study below to compare VCI to SAE and hierarchical autoencoder (HAE) with the same neural architecture.

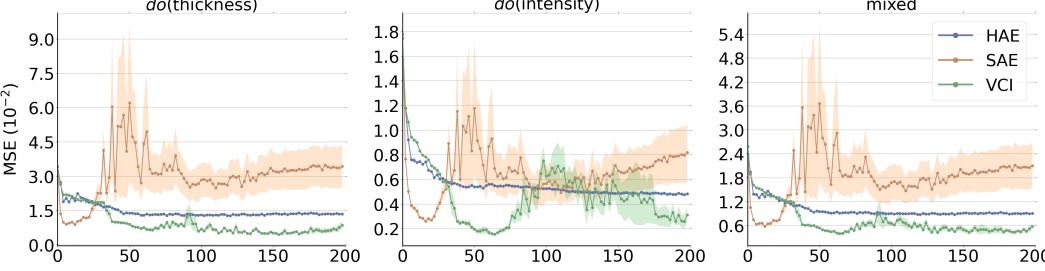

Figure 3: Ablation Study: the error of counterfactual prediction across epochs during the training of HAE (VCI without counterfactual supervision and latent divergence), SAE (VCI without latent divergence) and VCI over five independent runs. Note that VCI with latent divergence but without counterfactual supervision does not make logical sense, but for the completeness of the ablation study, we present the results for such setting in Appendix N.2.

As can be seen in Figure 3, incorporating counterfactual supervision alone could greatly accelerate optimization in the initial epochs of training, and achieves an ideal model state in the early stage. This is the same phenomenon we observed in the optimization on single-cell perturbation datasets with

respect to $\bar{R}^2$. However, although both SAE and VCI beat HAE convincingly in terms of best result, SAE is largely unstable in the long-run without the proposed latent divergence restriction, and the full VCI optimization scheme consistently achieves better performance in terms of best result and final result. To further help the readers understand how and why the latent divergence term stabilizes VCI training, we include a discussion below along with some illustrations on $do$(intensity) in Figure 4.

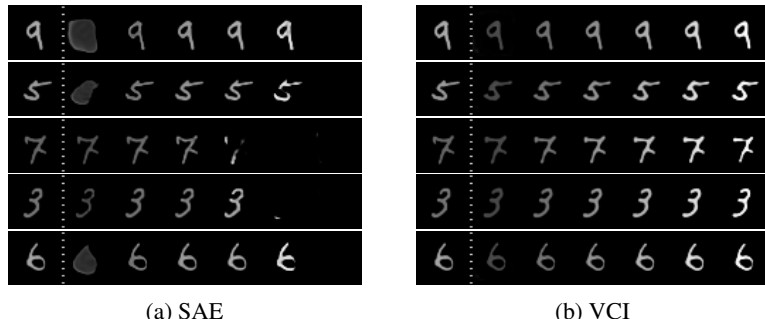

(a) SAE                          (b) VCI

Figure 4: Illustration of the latent divergence term's long-term impact. Samples are drawn from the last epoch of SAE and VCI on $do$(intensity). The left most image of each set is the original image. **More sampled results from VCI on $do$(thickness) and $do$(intensity) can be found in Figure 12. Results from intervening digit i.e. $do$(digit) can be found in Figure 13.**

In a GAN training scheme, the discriminator is a neural model that could often struggle with out-of-distribution evaluations. As can be seen from the generated samples by SAE in Figure 4, when the intervention labels are close to the original label, the counterfactual generations are reasonable. For intervention labels that are significantly different from the original, the images become nonsensical. This is because the combination of an intervention label that is far from the original and an image similar to the original is highly out-of-distribution and rarely present in the observed dataset. In other words, the discriminator has never encountered an image with the same digit and style as the original having such extreme intensity variations, so whatever the generator produces, the discriminator cannot tell if it is real or fake due to the lack of reference. This results in the generator exploiting the neural discriminator in these out-of-distribution scenarios in the long run. However, the latent divergence term prevents the generator from producing these nonsensical images because, otherwise, the encoder would not be able to recover a latent distribution close to that of the original image. **An evaluation of latent disentanglement under the latent divergence term can be found in Appendix N.3.**

### 3.3 CELEBA-HQ

For any real-world dataset where the counterfactual truth is not available, there is not a definitive metric to evaluate how good the counterfactual constructions really are. For that reason, facial imaging datasets are prevailing benchmarks for examining counterfactual goodness because even in the lack of a quantitative metric, human can judge the quality of counterfactuals without the need for any domain knowledge. Same as Monteiro et al. (2023), we use the CelebA-HQ (Karras et al., 2017) dataset on 64×64 resolution to evaluate the model's capability of counterfactual constructions on two factors – smiling and glasses. Some sampled results from our model are shown in Figure 5.

While MSE of the counterfactual construction cannot be measured, readers can empirically compare ours to prior work such as CHVAE (Monteiro et al., 2023) which serves as the backbone for the state-of-the-art model in high fidelity counterfactual image generation MED (Ribeiro et al., 2023). Aside from compiling counterfactual credibility across all generation tasks, ours show a strong capability of disentangling factual treatment from observed images in counterfactual construction, which can be particularly observed in the task of "removing glasses" as shown in Figure 5d. Note that contrary to prior works, we construct and supervise counterfactual outcomes during training, and the reconstruction and counterfactual construction share the same decoding mechanism. Hence, "reconstruction" is really counterfactual construction with the factual label. This can be most clearly observed in the third set of comparison in line 2 of Figure 5a, where the original image has a label of "not smiling" yet the person is really in-between smiling and not smiling. In this case, the "reconstruction" is really a counterfactual construction with treatment abiding to the original label.

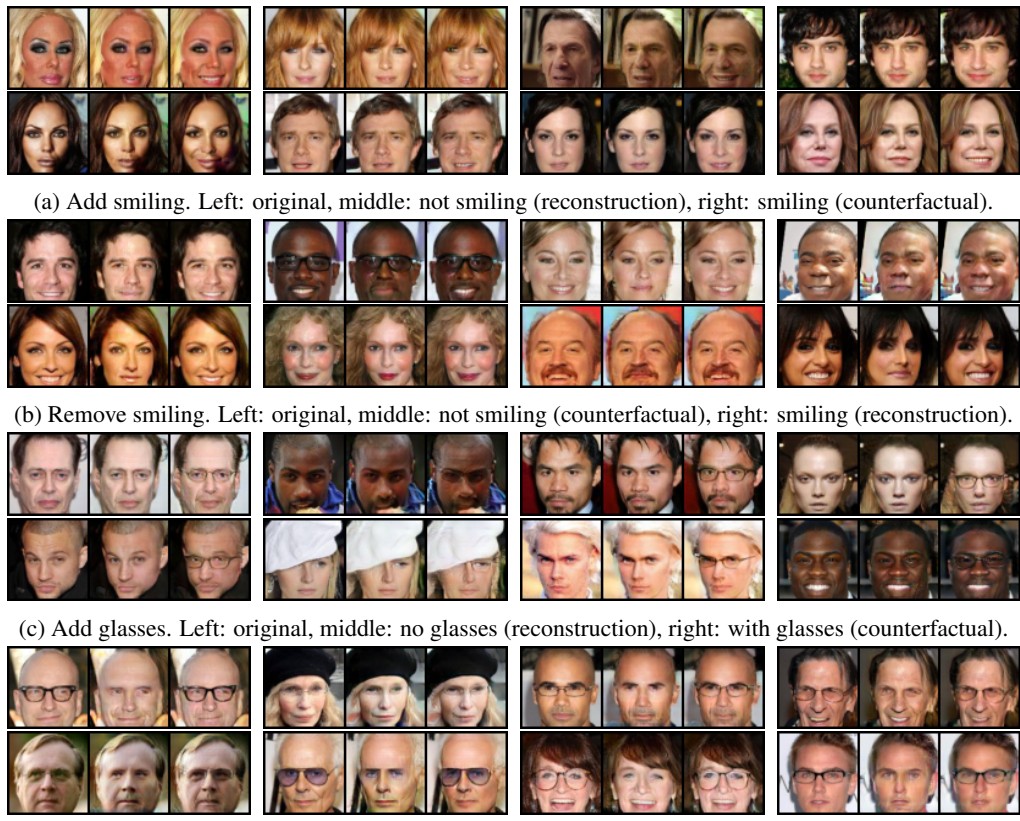

(a) Add smiling. Left: original, middle: not smiling (reconstruction), right: smiling (counterfactual).

(b) Remove smiling. Left: original, middle: not smiling (counterfactual), right: smiling (reconstruction).

(c) Add glasses. Left: original, middle: no glasses (reconstruction), right: with glasses (counterfactual).

(d) Remove glasses. Left: original, middle: no glasses (counterfactual), right: with glasses (reconstruction).

Figure 5: Results on the test set of CelebA-HQ.

Lastly, we include a discussion here regarding counterfactual fairness. Note that adding glasses is a relatively easy engineering task, but relatively hard deep learning task on CelebA-HQ due to the small sample size of images with glasses. As can be seen from Figure 5c, our model learned to not only attach glasses at the right position, but also attach glasses according to a person's style and context. However, readers may naturally wonder if this could induce any fairness issue, such as attaching certain type of glasses more frequently on certain demographic. We note that our framework can reduce spurious or unintended relations between given factors, as long as practitioners measure these factors during data collection and include them as observed labels for the composed datasets, since the learnt latent representations of our framework are encouraged to be disentangled from the observed factors. In the case of CelebA, style of glasses is unmeasured, and the model cannot reasonably tell if attaching different types of glasses on different demographics is unintended or not.

## 4    CONCLUSION

In this work, we introduced a variational Bayesian causal inference framework for estimating high-dimensional counterfactual outcomes, as well as consequent robust marginal estimators. With this framework, treatment characteristics and individuality in the predicted outcomes can be explicitly balanced and optimized, and learnt latent representations are disentangled from the treatments. As for limitations, the theoretical results of this work are established upon the common causal assumption that there exists no unobserved confounders, through which causal effects are identifiable. Besides, when multiple treatments are concerned, this work does not explicitly address the situation where a causal relation exists between treatments. Thus, a straightforward extension of this work could be to incorporate explicit modeling of the causal relation among treatment variables and update the descendants of given counterfactual treatment, similar to Pawlowski et al. (2020), before feeding them into our counterfactual outcome construction framework.

AUTHOR CONTRIBUTIONS

Yulun Wu developed the proposed framework, conducted experiments on single-cell perturbation datasets, Morpho-MNIST and CelebA-HQ, and wrote the manuscript; Louie McConnell conducted experiments on Morpho-MNIST; Claudia Iriondo reviewed related work and edited the manuscript.

ACKNOWLEDGMENTS

The authors thank Cian Eastwood, Toru Shirakawa, Pieter Abbeel, Jason Hartford, Dominik Janzing, Ivana Malenica, Ahmed M. Alaa, and David E. Heckerman for the insightful discussions. The authors thank Layne C. Price, Zichen Wang, Robert A. Barton, Vassilis N. Ioannidis, and George Karypis for their feedback in the early development.

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

## A    INTRODUCTION ON LATENT DISENTANGLEMENT

The study of latent disentanglement in deep causal modeling has mainly focused on two areas. Most prior works lie in the first area that is causal representation learning, in which researchers seek to learn latent variables corresponding to true causal factors and identify the structure of their causal relations (Yang et al., 2021; Hälvä et al., 2021; Gresele et al., 2021; Lachapelle et al., 2022; Komanduri et al., 2023), predominantly leveraging recent advances in non-linear independent component analysis (ICA) (Hyvarinen et al., 2019; Khemakhem et al., 2020) and identify causal structure up to Markov equivalence (Spirtes et al., 2000) or graph isomorphism under rather heavy assumptions. The second area and the area this work is concerned of is counterfactual generative modeling, in which the focus is to conduct disentangled exogenous noise abduction that aids counterfactual outcome generation (Lotfollahi et al., 2021; Shen et al., 2022). In this context, the goal is to acquire latent representations disentangled from the observed treatment to aid the correct identification of the causal effect of counterfactual treatments.

## B    RELATED WORK

Key prior works in deep causal generative modeling fall primarily into three categories: variational-based methods **CEVAE** (Louizos et al., 2017), **DeepSCM** (Pawlowski et al., 2020), **Diff-SCM** (Sanchez & Tsaftaris, 2022), **CHVAE** (Monteiro et al., 2023), **MED** (Ribeiro et al., 2023); adversarial-based methods **CausalGAN** (Kocaoglu et al., 2017), **GANITE** (Yoon et al., 2018); hybrid method **DEAR** (Shen et al., 2022) as well as our framework **VCI**. A comparative analysis of ours against these related works can be found in Table 3.

Table 3: A comparative analysis of related work. "Variational Objective Distribution" describes the objective that the ELBOs are derived for in a variational-based method: "Joint" indicates joint distributions such as $p(y, x, t)$ or $p(y, z)$ (note that different works could have different notations, here we use $y$, $x$, $t$, $z$ to represent outcome, covariates, treatment, latent respectively), and "Interventional" indicates conditional outcome distribution $p(y|x, t)$ or $p(y|t)$. "Marginal Distribution Alignment" describes whether the stochastic optimization objective involves mechanisms that match the distribution of outcome constructions with some learnt marginal outcome distribution.

| | Type | Variational Objective Distribution | Hierarchical Model Structure | Marginal Distribution Alignment | Causal Discovery | End-to-End Counterfactual Supervision | Latent Disentanglement |
|---|---|---|---|---|---|---|---|
| CEVAE | Variational | Joint | ✗ | ✗ | ✓ | ✗ | ✗ |
| CausalGAN | Adversarial | – | ✗ | ✓ | ✗ | ✗ | ✗ |
| GANITE | Adversarial | – | ✗ | ✓ | ✗ | ✗ | ✗ |
| CPA | Adversarial | – | ✗ | ✗ | ✗ | ✗ | ✓ |
| DEAR | Variational + Adversarial | Joint | ✗ | ✗ | ✓ | ✗ | * |
| DeepSCM | Variational | Interventional | ✗ | ✗ | ✗ | ✗ | ✗ |
| Diff-SCM | Variational | Interventional | ✓ | ✗ | ✗ | ✗ | ✗ |
| CHVAE | Variational | Interventional | ✓ | ✗ | ✗ | ✗ | ✗ |
| MED | Variational | Interventional | ✓ | ✓ | ✗ | ✗ | ✗ |
| VCI | Variational + Adversarial | Counterfactual | ✓ | ✓ | ✗ | ✓ | ✓ |

*DEAR learns restrictive representation for latent $z$ ($z$ is disentangled from treatment $t$ when encoding $t$) but does not learn consistent representation for $z$ ($t$ is not disentangled from $z$ when encoding $z$).

**CEVAE** uses variational autoencoders (VAEs) to infer latent confounders and estimate individual treatment effects (ITE). **DeepSCM** integrates deep learning into SCMs using normalizing flows and variational inference for counterfactual inference. **Diff-SCM** leverages generative diffusion models, using forward and reverse diffusion processes guided by anti-causal predictors to generate counterfactuals. **CHVAE** extends VDVAE to a conditional model, performing abduction by encoding images and parent attributes, and includes a penalty for counterfactual conditioning. **MED** uses hierarchical VAEs with deep causal mechanisms, inferring exogenous noise by conditioning on observed data and parent variables. **GANITE** uses a counterfactual generator to create proxies for

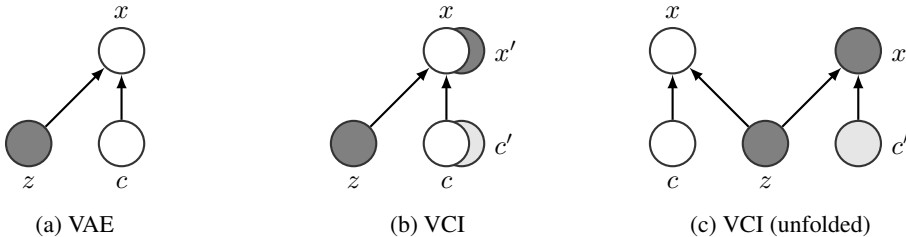

(a) VAE           (b) VCI           (c) VCI (unfolded)

Figure 6: The data generating formulation of conditional generative modeling under VAE formulation and counterfactual generative modeling under VCI formulation using standard VAE notations. $p(x|z,c) = p(x'|z,c')$ for $c = c'$. White nodes are observed, light grey nodes are assigned during training and inference, dark grey nodes are unobserved.

Table 4: The objective and ELBO for conditional generative modeling under the VAE formulation and counterfactual generative modeling under the VCI formulation, using standard VAE notations. As can be seen, the difference in motivations between the two objectives manifest themselves in the derived ELBOs in some meaningful and intuitive ways: VAE does not have the motivation to maximize counterfactual outcome likelihood, hence no such term $\log[\hat{p}(x'_{\theta,\phi}|c')]$ to conduct supervision of counterfactuals; VCI does not have the motivation to denoise or to generate samples from pure noise duing inference time, hence no such term $D_{\mathrm{KL}}[q_\phi(z|x,c) \| p_\theta(z|c)]$ to bound latent distribution on some marginal prior.

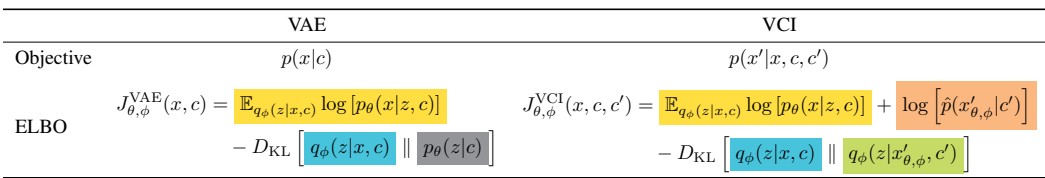

| | VAE | VCI |
|---|---|---|
| Objective | $p(x\|c)$ | $p(x'\|x,c,c')$ |
| ELBO | $J^{\mathrm{VAE}}_{\theta,\phi}(x,c) = \boxed{\mathbb{E}_{q_\phi(z\|x,c)} \log [p_\theta(x\|z,c)]}$ $- D_{\mathrm{KL}}\left[ \boxed{q_\phi(z\|x,c)} \| \boxed{p_\theta(z\|c)} \right]$ | $J^{\mathrm{VCI}}_{\theta,\phi}(x,c,c') = \boxed{\mathbb{E}_{q_\phi(z\|x,c)} \log [p_\theta(x\|z,c)]} + \boxed{\log\left[ \hat{p}(x'_{\theta,\phi}\|c') \right]}$ $- D_{\mathrm{KL}}\left[ \boxed{q_\phi(z\|x,c)} \| \boxed{q_\phi(z\|x'_{\theta,\phi},c')} \right]$ |

unobserved outcomes and an ITE generator to estimate potential outcomes. **DEAR** combines a VAE-like generative loss with a supervised loss in a bidirectional generative model, trained with a GAN algorithm requiring knowledge of the causal graph structure and extra labels for latent factors.

As illustrated in Section 2 and Appendix C, the primary contribution of our framework **VCI** is the revolution of VAE formulation which leads to a brand new stochastic optimization scheme that optimizes the ELBO of an actual counterfactual objective. This brings about end-to-end counterfactual supervision and exogenous noise disentanglement as presented in Table 3.

## C COMPARISON OF CONDITIONAL VAE AND VCI

For readers that are more familiar with deep generative modeling than causal inference, we provide a straightforward comparison between conditional VAE and VCI (fundamentally a comparison between conditional generative modeling and counterfactual generative modeling) in this section using VAE's notations. For data $x$ and condition $c$, the differences in formulations and objectives are shown in Figure 6 and Table 4. In causal inference, counterfactual outcome is an individual-level concept – for a given individual that we observed outcome/data $x$ under treatment/condition $c$, what would their outcome have been if they had received treatment/condition $c'$ instead? This is a "would have" question, meaning that we seek to find out what the alternative outcome $x'$ would be in a parallel world where everything in the state of the universe remained the same, except that the treatment $c'$ (and its impact) was different. With this in mind, it is not hard to see why the conditional generative modeling formulation is unorthodox for counterfactual generative modeling – the ELBO serves to optimize the marginal-level likelihood $p(x|c)$, which is interventional (rung 2) and not counterfactual (rung 3) (Pearl, 1995), and the learnt model generates samples towards the marginal-level distribution $p(x|c)$ during inference time, and the question being asked here is "what will an outcome $x$ be under condition $c$?". This is a "will" question – it is generating *any* outcome under condition $c$, not given a specific individual, not given a specific state of the universe. Therefore, contrary to prior works in HVAEs and diffusion models (Sanchez & Tsaftaris, 2022; Monteiro et al., 2023; Ribeiro et al.,

2023) which focus on model design adaptations to make the conditional VAE formulation work in counterfactual generative modeling, we first derive the orthodox formulation and objective for counterfactual generative modeling, then make the necessary model designs afterwards based on the accordingly derived ELBO.

## C.1 DIFFUSION MODELS

Due to the popularity of the diffusion models (Ho et al., 2020), we feel the necessity to include a discussion here specifically about its compatibility with counterfactual generative modeling. It is very important to note that, although diffusion models are the state-of-the-art in generative and conditional generative modeling, it has not been shown that it has better capability in counterfactual generative modeling than ordinary HVAEs with learnable encoder (Monteiro et al., 2023; Ribeiro et al., 2023), and the reason is straightforward once we truly understand the fundamental incompatibility between the goal of counterfactual inference and the diffusion mechanism. Counterfactual inference entails abducting and preserving the exogenous noise in the original outcome; diffusion models, on the other hand, entail diffusing and denoising the original image – it is a mechanism that inherently does not respect the consistency assumption (Assumption 2) and fundamentally contradicts exogenous noise abduction. Even if certain prior work such as Sanchez & Tsaftaris (2022) uses the learnt noise model to replace random noise in inference time, it is still performing the illogical operation of attempting to construct counterfactual from a noisy version of the original outcome, in which the exogenous noise has already been discarded. And this is on top of the fact that diffusion models, as models under the VAE formulation, are working towards the wrong interventional objective as discussed above. Prior works that attempted at utilizing diffusion models in counterfactual generative modeling (Dash et al., 2022; Sanchez & Tsaftaris, 2022; Komanduri et al., 2024) clearly exhibited this flaw: as can be seen from the CelebA results in Dash et al. (2022) and the MNIST results in Komanduri et al. (2024), exogenous noise/individuality has been largely discarded in counterfactual generation – because the diffusion model is doing what it is intended to do: diffusing exogenous and generating samples that fit seeminglessly into the conditional/marginal likelihood $p(x|c)$; Sanchez & Tsaftaris (2022) on the other hand, hardly showed any evidence that the model is capable of preserving individuality – MNIST was the only benchmark in their experiments and only intervening digit was performed, which does not have measurable counterfactual truth as intervening thickness and intensity in Morpho-MNIST. In general, the "interventions" performed in Sanchez & Tsaftaris (2022) all drastically modified the objects in the original images and makes it quite impossible to tell if exogenous noise are preserved either quantitatively or qualitatively. When one drastically modifies the objects, the problem moves very far away from the consistency assumption, and it becomes really questionable whether the results under such "intervention" can even be defined as counterfactuals or not. We want to clarify that we think Dash et al. (2022), Sanchez & Tsaftaris (2022) and Komanduri et al. (2024) are meaningful and well-written works that conducted the important advancement of experimenting diffusion models in counterfactual generative modeling, however, diffusion models have not shown the capability of exogenous noise abduction on the level of state-of-the-art HVAEs (Monteiro et al., 2023; Ribeiro et al., 2023) as it stands.

To give the readers a sense of how much difference it makes to use the VCI framework instead of conditional diffusion models for counterfactual generative modeling, we presented the visual results on CelebA in Figure 5 and the results on "intervening" digits of MNIST in Figure 13 for readers to compare to aforementioned prior arts.

## D COMPARISON OF TRADITIONAL CAUSAL FORMULATION AND VCI FORMULATION

We note that we formulate counterfactuals $Y'$ and $T'$ as separate variables for the cleanliness of graphical model and notation simplicity in variational inference, with the goal to bridge readers with variational inference background and causal inference background. In essence, it is no different from the traditional causal formulation where the $\mathrm{do}()$ operation is directly imposed on factual variable $T$, as $T'$ is just a dummy variable with no observations. Denote $Y_{\mathrm{do}(T=t')}$ as $Y_{t'}$, see Figure 7 for a demonstration of our formulation in traditional notations.

Then under the ignorability assumption (Assumption 1, note that there is no $\epsilon_Y$ and $\epsilon'_Y$ in the traditional SCM formulation as there is only one copy of exogenous noise $U_Y$ injected to factual and

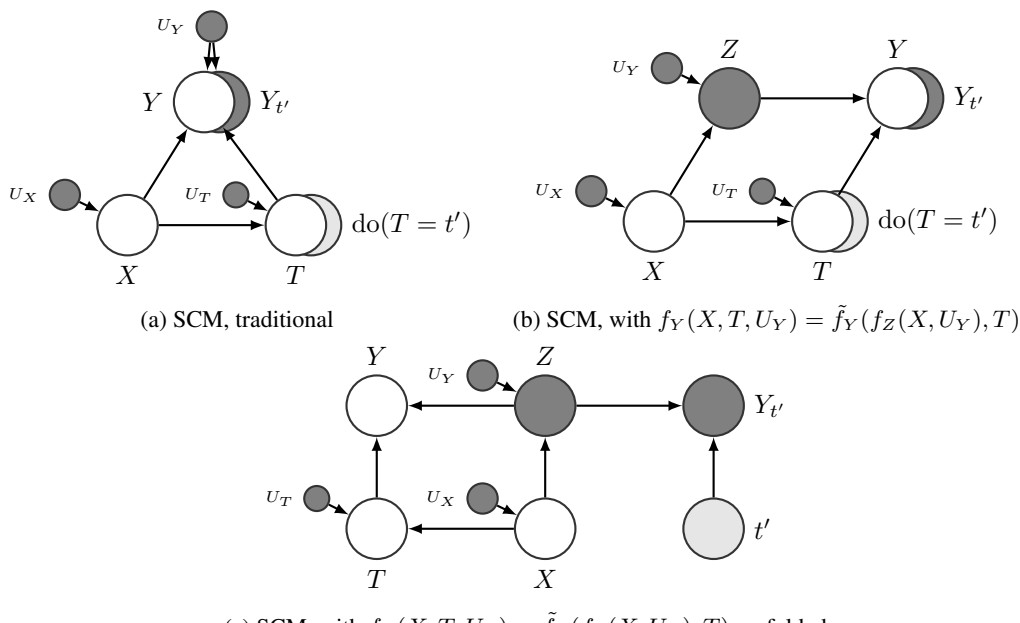

(a) SCM, traditional

(b) SCM, with $f_Y(X, T, U_Y) = \tilde{f}_Y(f_Z(X, U_Y), T)$

(c) SCM, with $f_Y(X, T, U_Y) = \tilde{f}_Y(f_Z(X, U_Y), T)$, unfolded

Figure 7: The evolution of our formulation from traditional SCM with observed triplets $(X, T, Y)$. Figure 7b is simply extending the production function towards $Y$ with an intermediate variable $Z$ representing two of its inputs (when $f_Z(X, U_Y) = (X, U_Y)$, $\tilde{f}_Y$ is just $f_Y$). Then, Figure 7c is just the unfolded graphical model of Figure 7b, as the $\mathrm{do}()$ operation replaces $T$ in the production function of $Y$ with $t'$, i.e. $Y_{t'} = f_Y(X, t', U_Y)$.

counterfactual outcome, but in general if there exists $\epsilon_Y$ and $\epsilon'_Y$ independent of $U_Y$, the following result holds as long as $\epsilon_Y, \epsilon'_Y \perp\!\!\!\perp U_T$), we have the following version of Theorem 1:

**Corollary 2.** *Suppose a collection of random variables follows a causal structure defined by the Bayesian network in Figure 7b. Then $\log[p(Y_{t'}|Y, X, T, t')]$ has the following variational lower bound:*

$$\log\left[p(Y_{t'}|Y, X, T, t')\right] \geq \mathbb{E}_{p(Z|Y, T, X)} \log\left[p(Y|Z, T)\right] - D\left[p(Y|X, T) \,\|\, p(Y_{t'}|X, t')\right]$$
$$- D_{\mathrm{KL}}\left[p(Z|Y, T, X) \,\|\, p(Z|Y_{t'}, t', X)\right] \quad (10)$$

*where $D[p \,\|\, q] = \log p - \log q$.*

Now we define dummy variable

$$T' = f_T(X, U'_T), \quad (11)$$

with $U'_T$ satisfying the ignorability assumption (Remark 1), and

$$Y' = f_Y(X, T', U_Y) = \tilde{f}_Y(f_Z(X, U_Y), T'). \quad (12)$$

Then, $Y'_{\mathrm{do}(T'=t')} = f_Y(X, t', U_Y) = Y_{t'}$. Hence, we have

$$p(Y_{t'}|Y, X, T, t') = p(Y'|Y, X, T, do(T'=t')) = p(Y'|Y, X, T, T'=t') \quad (13)$$

for any $t'$. Therefore, Figure 1a and Theorem 1 is in essence just Figure 7c and Corollary 2 with different notations and variable definitions.

# E IMPLICIT COUNTERFACTUAL SUPERVISION

We demonstrated in Appendix C that the marginal-level observed likelihood is unorthodox for counterfactual generative modeling and the individual-level likelihood $\log[p(Y'|Y, X, T, T')]$ is better motivated for this task. However, for researchers who insist on working towards the traditional likelihood objective $\log[p(Y|X, T)]$ composed only of observed variables, there is an approach to conduct counterfactual supervision implicitly – to view $Y'$ as a latent variable restricted on the prior $p(Y'|X, T')$ and view counterfactual construction task as a latent recognition task:

**Proposition 3.** $J(O) = \log[p(Y|X, T)]$ *has the following variational lower bound:*

$$J(O) \geq \mathbb{E}_{p(Z|Y,T,X)} \log [p(Y|Z,T)] - \mathbb{E}_{p(Z|Y,T,X)} D_{\mathrm{KL}} [p(Y'|Z,T') \parallel p(Y'|X,T')]$$
$$- D_{\mathrm{KL}} [p(Z|Y,T,X)p(Y'|Z,T') \parallel p(Z|Y',T',X)p(Y'|Z,T')] . \tag{14}$$

This induces the following ELBO in stochastic optimization:

$$J_{\theta,\phi}(o) = \mathbb{E}_{q_\phi(z|y,t,x)} \log [p_\theta(y|z,t)] - \mathbb{E}_{q_\phi(z|y,t,x)} D_{\mathrm{KL}} [p_\theta(y'|z,t') \parallel \hat{p}(y'|x,t')]$$
$$- D_{\mathrm{KL}} [q_\phi(z|y,t,x)p_\theta(y'|z,t') \parallel q_\phi(z|y',t',x)p_\theta(y'|z,t')] . \tag{15}$$

Equation 15 avoids estimating $Y'$ with a single sample as in the objective (Equation 3) derived from the explicit ELBO (Equation 2). However, it could be a lot more sampling inefficient as the latent divergence term requires Monte-Carlo samples from both $Z$ and $Y'$ to be estimated.

# F    DEFINITION OF ORACLE CONSISTENCY, ORACLE RESTRICTIVENESS, AND DISENTANGLEMENT

Let $\mathcal{H} = \{(\tilde{p}, \tilde{q}) \mid \tilde{p} : \mathcal{S} \to [0,1]^{\Sigma_\mathcal{V}}, \ \tilde{q} : \mathcal{V} \times \mathcal{X} \to [0,1]^{\Sigma_\mathcal{S}}\}$ be a hypothesis class of stochastic models $(\tilde{p}, \tilde{q})$. The input space of $\tilde{p}$ is a measurable space $\mathcal{S}$ where a feature vector $s \in \mathcal{S}$ is drawn; the input space of $\tilde{q}$ is the product space of $\mathcal{V}$ and $\mathcal{X}$ where a vector of interest $v \in \mathcal{V}$ and some auxiliary attributes $x \in \mathcal{X}$ are drawn. The outputs of $\tilde{p}$ and $\tilde{q}$ are probability measures on $\mathcal{V}$ and $\mathcal{S}$ respectively. Let the true model $(p^*, q^*)$ be the true conditional distributions $(p_0(V|s), p_0(S|v, x))$ of random variables $V$ and $S$. The following definitions provide criteria under which two parts $S_I$ and $S_{\setminus I}$ of feature variable $S$ separated by an index set $I$ are disentangled by a model.

**Definition 1** (Oracle Consistency). *Consider* $s = (s_I, s_{\setminus I})$ *and* $s' = (s_I, s'_{\setminus I})$ *where*

$$x \sim p(X) \tag{16}$$
$$s_I \sim p(S_I|x) \tag{17}$$
$$s_{\setminus I}, s'_{\setminus I} \overset{\mathrm{iid}}{\sim} p(S_{\setminus I}|s_I, x) \tag{18}$$

*We say that* $\tilde{q}$ *is consistent with* $S_I$ *if* $p$ *is the true distribution* $p_0$ *and*

$$\mathbb{E}_{p_0, p^*} \|\tilde{q}_I \circ p^*(s; x) - \tilde{q}_I \circ p^*(s'; x)\| = 0, \tag{19}$$

*where* $\tilde{q}_I(s_I|v, x) = \int_{s_{\setminus I}} \tilde{q}(s_I, s_{\setminus I}|v, x)$. *More generally, we say that* $(\tilde{p}, \tilde{q})$ *is consistent with* $S_I$ *under* $p$ *if*

$$\mathbb{E}_{p, \tilde{p}} \|\tilde{q}_I \circ \tilde{p}(s; x) - \tilde{q}_I \circ \tilde{p}(s'; x)\| = 0. \tag{20}$$

As discussed in Shu et al. (2019), oracle consistency (or encoder consistency) states that for any fixed choice of $S_I$, resampling $S_{\setminus I}$ should not affect the oracle's measurement of $S_I$. A more general notion of oracle consistency is defined in addition, as it is valuable in real-world optimization to evaluate the estimating model's consistency under its own oracle $\tilde{q} \circ \tilde{p}$.

In contrast, restrictiveness states that for any fixed choice of $S_{\setminus I}$, resampling $S_I$ should not affect the oracle's measurement of $S_{\setminus I}$:

**Definition 2** (Oracle Restrictiveness). *Consider* $s = (s_I, s_{\setminus I})$ *and* $s' = (s'_I, s_{\setminus I})$ *where*

$$x \sim p(X) \tag{21}$$
$$s_{\setminus I} \sim p(S_{\setminus I}|x) \tag{22}$$
$$s_I, s'_I \overset{\mathrm{iid}}{\sim} p(S_I|s_{\setminus I}, x) \tag{23}$$

*We say that* $\tilde{q}$ *is restricted to* $S_I$ *if* $p$ *is the true distribution* $p_0$ *and*

$$\mathbb{E}_{p_0, p^*} \|\tilde{q}_{\setminus I} \circ p^*(s; x) - \tilde{q}_{\setminus I} \circ p^*(s'; x)\| = 0, \tag{24}$$

*where* $\tilde{q}_{\setminus I}(s_{\setminus I}|v, x) = \int_{s_I} \tilde{q}(s_I, s_{\setminus I}|v, x)$. *More generally, we say that* $(\tilde{p}, \tilde{q})$ *is restricted to* $S_I$ *under* $p$ *if*

$$\mathbb{E}_{p, \tilde{p}} \|\tilde{q}_{\setminus I} \circ \tilde{p}(s; x) - \tilde{q}_{\setminus I} \circ \tilde{p}(s'; x)\| = 0. \tag{25}$$

Notice that the oracle consistency and restrictiveness of $S_I$ is equivalent to the oracle restrictiveness and consistency of $S_{\setminus I}$, we have the following definition of disentanglement:

**Definition 3** (Disentanglement). *We say that $\tilde{q}$ disentangles $S_I$ and $S_{\setminus I}$ if $\tilde{q}$ is consistent with and restricted to $S_I$. More generally, we say that $(\tilde{p}, \tilde{q})$ disentangles $S_I$ and $S_{\setminus I}$ under $p$ if $(\tilde{p}, \tilde{q})$ is consistent with and restricted to $S_I$ under $p$.*

## G  ROBUST MARGINAL EFFECT ESTIMATION

By Van der Vaart (2000), a sequence of estimators $\hat{\Psi}_n$ is asymptotically efficient if $\sqrt{n}(\hat{\Psi}_n - \Psi(p)) = 1/\sqrt{n} \sum_{k=1}^{n} \tilde{\psi}_p(W_k) + o_p(1)$ where $\tilde{\psi}_p$ is the efficient influence function of $\Psi(p)$ and $W_k \sim p(W)$. The theorem below gives this efficient influence function for the average treatment effect of the treated (ATT) $\Psi(p) = \mathbb{E}_p[Y'_{\mathrm{do}(T'=\alpha)}]$ and thus provides a construction of an asymptotically efficient regular estimator for $\Psi$:

**Theorem 2.** *Suppose $W : \Omega \to \mathcal{R}_W$ follows a causal structure defined by the Bayesian network in Figure 1, where the counterfactual conditional distribution $p(Y', T'|Z, X)$ is identical to that of its factual counterpart $p(Y, T|Z, X)$. Then $\Psi(p)$ has the following efficient influence function:*

$$\tilde{\psi}_p(W) = \frac{I(T = \alpha)}{p(T|X)}(Y - \mathbb{E}_p[Y|Z, T]) + \mathbb{E}_p[Y'|Z, T' = \alpha] - \Psi. \tag{26}$$

This leads to the construction of $\hat{\Psi}_n(o)$ in Equation 4 which is asymptotically efficient under some regularity conditions (Van Der Laan & Rubin, 2006), with the asymptotic distribution given by

$$\sqrt{n}(\hat{\Psi}_n - \Psi(p)) \xrightarrow{d} \mathcal{N}(0, E_p[\tilde{\psi}_p(W)\tilde{\psi}_p(W)^T]) \tag{27}$$

where the variance can be estimated empirically in practice and confidence bounds can be constructed accordingly. The robust estimator within the VCI framework can also be extended to estimating covariate-specific marginal treatment effect $\Xi(p) = \mathbb{E}_p[Y'_{\mathrm{do}(X=c,T'=\alpha)}]$ for a given covariate $c$ of interest:

**Corollary 3.** *Under the same settings as Theorem 2, $\Xi(p)$ has the following efficient influence function:*

$$\tilde{\xi}_p(W) = \frac{I(X = c)}{p(X)}\tilde{\psi}_p(W). \tag{28}$$

Note that when $p(X)$ is estimated empirically, the asymptotically efficient estimator for $\Xi(p)$:

$$\hat{\Xi}_n(o) = \frac{1}{n} \sum_{i=1}^{n} \frac{I(x_k = c)}{\hat{p}(x_k)} \left\{ \frac{I(t_k = \alpha)}{\hat{e}(t_k|x_k)} \cdot y_k + \left(1 - \frac{I(t_k = \alpha)}{\hat{e}(t_k|x_k)}\right) \cdot \mathbb{E}_{p_\theta}[Y'|z_{k,\phi}, T' = \alpha] \right\} \tag{29}$$

$$= \frac{1}{n_c} \sum_{i=1}^{n_c} \left\{ \frac{I(t_{k_i} = \alpha)}{\hat{e}(t_{k_i}|x_{k_i})} \cdot y_{k_i} + \left(1 - \frac{I(t_{k_i} = \alpha)}{\hat{e}(t_{k_i}|x_{k_i})}\right) \cdot \mathbb{E}_{p_\theta}[Y'|z_{k_i,\phi}, T' = \alpha] \right\} \tag{30}$$

is simply an application of Equation 4 on the set of observations with covariates $c$, where $\{k_1, k_2, \ldots, k_{n_c}\}$ are indices such that $x_{k_i} = c$.

## H  FORMAL DEFINITION OF COMMON CAUSAL ASSUMPTIONS UNDER THE VCI FORMULATION

**Assumption 1** (Ignorability). *There is no unobserved confounders, i.e. $\tilde{U}_Y \perp\!\!\!\perp U_T$, where $\tilde{U}_Y = (U_Y, \epsilon_Y, \epsilon'_Y)$.*

Note that this assumption can be extended to the counterfactual treatment variable $T'$ w.o.l.g.:

**Remark 1** (Ignorability). *Note that $T'$ in our formulation is a dummy variable with no observations, and we only care about the outcome distribution of $Y'$ under assignment $do(T' = t')$ for any given $t'$. Hence, the ignorability $\tilde{U}_Y \perp\!\!\!\perp U'_T$, and in addition, $U_X \perp\!\!\!\perp U'_T$, can be assumed w.o.l.g. just so that the outcome distributions under $do(T' = t')$ reduces to that under $T' = t'$ conditioned on $X$ or $Z$ for notation simplicity, without violating any actual mechanism on the observed data.*

The ignorability assumption is universally assumed and entailed in our formulation in Figure 1.

**Assumption 2** (Consistency). *An individual's observed outcome under a treatment is the same as its potential outcome under that treatment, i.e. $Y = Y'_{\text{do}(T'=T)}$, where $Y'_{\text{do}(T'=T)} = f_Y(Z, T, \epsilon'_Y)$ is yielded by replacing the production equation for $T'$ with $T' = T$.*

The consistency assumption, commonly stated in causal inference literature, is indeed an assumption and not just a direct consequence of the SCM under our formulation, since $\epsilon_Y$ and $\epsilon'_Y$ are two copies of independent variables. In essence, it states that $\epsilon_Y = 0$ and there is only one copy of exogenous noise $U_Y$ injected to $Y$ and $Y'$:

**Remark 2** (Consistency). *Under the consistency assumption, $f_Y(z, t, \epsilon_Y)$ must be deterministic for any $z$ and $t$, otherwise we would have $Y \neq Y'_{\text{do}(T'=T)}$ since $f_Y(z, t, \epsilon_Y)$ and $f_Y(z, t, \epsilon'_Y)$ are independent for any $z$ and $t$. Hence there exists deterministic function $\tilde{f}_Y$ such that $\tilde{f}_Y(Z, T, 0) = f_Y(Z, T, \epsilon_Y)$. So w.o.l.g., we let $\epsilon_Y = 0$ under the consistency assumption.*

Note that this assumption does not result in a collapse of counterfactual inference to interventional inference in the traditional sense, as latent $Z$ is unobserved and entails further uncertainty $U_Y$ beyond the observed covariates $X$.

# I PROOF OF THEORETICAL RESULTS

## I.1 EVIDENCE LOWER BOUND

### I.1.1 PROOF OF THEOREM 1

*Proof.* By the d-separation (Pearl, 1988) of paths on the causal graph defined in Figure 1a, we have

$$\log\left[p(Y'|Y, X, T, T')\right] = \log \mathbb{E}_{p(Z|Y,T,X)}\left[p(Y'|Z, Y, X, T, T')\right] \tag{31}$$

$$\geq \mathbb{E}_{p(Z|Y,T,X)} \log\left[p(Y'|Z, Y, X, T, T')\right] \quad \text{(Jensen's inequality)} \tag{32}$$

$$= \mathbb{E}_{p(Z|Y,T,X)} \log \frac{p(Y', Z|Y, X, T, T')}{p(Z|Y, T, X)} \tag{33}$$

$$= \mathbb{E}_{p(Z|Y,T,X)} \log \frac{p(Y', Z, Y|X, T, T')}{p(Z|Y, T, X)p(Y|X, T)} \tag{34}$$

$$= \mathbb{E}_{p(Z|Y,T,X)} \log \frac{p(Y|Z, T)p(Z|Y', T', X)p(Y'|X, T')}{p(Z|Y, T, X)p(Y|X, T)} \tag{35}$$

$$= \mathbb{E}_{p(Z|Y,T,X)} \log\left[p(Y|Z, T)\right] - D_{\text{KL}}\left[p(Z|Y, T, X) \,\|\, p(Z|Y', T', X)\right]$$
$$- D\left[p(Y|X, T) \,\|\, p(Y'|X, T')\right]. \tag{36}$$

Note that the above steps still hold with an additional edge from $Z$ or $T$ to $X$ (but not both) if $U_Y$ or $U_T$ is dependent of $U_X$. $\qquad\square$

### I.1.2 PROOF OF PROPOSITION 3

*Proof.* By the d-separation (Pearl, 1988) of paths on the causal graph defined in Figure 1a, we have

$$\log\left[p(Y|X, T)\right] = \log\left[p(Y|X, T, T')\right] = \log \mathbb{E}_{p(Z,Y'|Y,X,T,T')} \frac{p(Y, Z, Y'|X, T, T')}{p(Z, Y'|Y, X, T, T')} \tag{37}$$

$$\geq \mathbb{E}_{p(Z,Y'|Y,X,T,T')} \log \frac{p(Y, Z, Y'|X, T, T')}{p(Z, Y'|Y, X, T, T')} \quad \text{(Jensen's inequality)} \tag{38}$$

$$= \mathbb{E}_{p(Z,Y'|Y,X,T,T')} \log \frac{p(Y'|X, T')p(Y, Z|Y', X, T, T')}{p(Z, Y'|Y, X, T, T')} \tag{39}$$

$$= \mathbb{E}_{p(Z,Y'|Y,X,T,T')} \log \frac{p(Y'|X, T')p(Z|Y', T', X)p(Y|Z, T)}{p(Z|Y, T, X)p(Y'|Z, T')} \tag{40}$$

$$= \mathbb{E}_{p(Z,Y'|Y,X,T,T')} \log \frac{p(Y'|X, T')p(Z|Y', T', X)p(Y'|Z, T')p(Y|Z, T)}{p(Y'|Z, T')p(Z|Y, T, X)p(Y'|Z, T')} \tag{41}$$

$$
\begin{aligned}
= & -\mathbb{E}_{p(Z|Y,T,X)} D_{\mathrm{KL}}\left[p(Y'|Z,T') \,\|\, p(Y'|X,T')\right] \\
& - D_{\mathrm{KL}}\left[p(Z|Y,T,X)p(Y'|Z,T') \,\|\, p(Z|Y',T',X)p(Y'|Z,T')\right] \\
& + \mathbb{E}_{p(Z|Y,T,X)} \log\left[p(Y|Z,T)\right].
\end{aligned}
\tag{42}
$$

Note that the above steps still hold with an additional edge from $Z$ or $T$ to $X$ (but not both) if $U_Y$ or $U_T$ is dependent of $U_X$. $\qquad\square$

### I.1.3  PROOF OF COROLLARY 2

*Proof.* Note that Theorem 1 only relies on the conditional dependency structure in Figure 1, and the only difference between the dependency structure of $(X,Z,T,T',Y,Y')$ in Figure 1 and that of $(X,Z,T,t',Y,Y_{t'})$ in Figure 7 is the dependency between $(X,T')$ and $(X,t')$. However, such difference does not change the conditional independence of latent posterior: $p(Z|Y,T,X,t') = p(Z|Y,T,X)$ and $p(Z|Y_{t'},t',X,T) = p(Z|Y_{t'},t',X)$, the conditional independence of outcome distribution $p(Y|Z,T,t') = p(Y|Z,T)$ and $p(Y_{t'}|Z,t',T) = p(Y_{t'}|Z,t')$, and the conditional independence of covariate-specific outcome distribution $p(Y|X,T,t') = p(Y|X,T)$ and $p(Y_{t'}|X,t',T) = p(Y_{t'}|X,t')$. Hence, the same proof for Theorem 1 applies to Corollary 2 by replacing $T'$ with $t'$ and $Y'$ with $Y_{t'}$. $\qquad\square$

## I.2  DISENTANGLED EXOGENOUS NOISE ABDUCTION

### I.2.1  PROOF OF LEMMA 1

*Proof.* Let $S_I = Z$ and $S_{\setminus I} = T$. On one hand,

$$
\mathbb{E}_{p_0,p*}\|\tilde{q}_I \circ p^*((s_I,s_{\setminus I});x) - \tilde{q}_I \circ p^*((s_I,s'_{\setminus I});x)\|
$$

$$
= \mathbb{E}_{p_0(x),p_0(s_I,s_{\setminus I},s'_{\setminus I}|x),p_0(v|s_I,s_{\setminus I}),p_0(v'|s_I,s'_{\setminus I})}\|\tilde{q}_I(v,x) - \tilde{q}_I(v',x)\|
\tag{43}
$$

$$
= \mathbb{E}_{p_0(x)p_0(z,t,t'|x),p_0(y|z,t),p_0(y'|z,t')}\|\tilde{q}_I(y,t,x) - \tilde{q}_I(y',t',x)\|
\tag{44}
$$

$$
= \mathbb{E}_{p_0(x,t,t',y,y')}\|q(\cdot|y,t,x) - q(\cdot|y',t',x)\|
\tag{45}
$$

$$
\leq \mathbb{E}_{p_0(d)}\sqrt{\frac{1}{2}D_{\mathrm{KL}}\left[q(\cdot|y,t,x) \,\|\, q(\cdot|y',t',x)\right]} \quad \text{(Pinsker's inequality)}
\tag{46}
$$

$$
= 0,
\tag{47}
$$

since Equation 5 implies that $D_{\mathrm{KL}}[q(\cdot|y,t,x) \,\|\, q(\cdot|y',t',x)] = 0$ a.e. with respect to $p_0$. Hence $\tilde{q}$ is consistent with $Z$. On the other hand, $\tilde{q}$ is restricted to $Z$ by design:

$$
\mathbb{E}_{p_0,p*}\|\tilde{q}_{\setminus I} \circ p^*((s_I,s_{\setminus I});x) - \tilde{q}_{\setminus I} \circ p^*((s'_I,s_{\setminus I});x)\|
$$

$$
= \mathbb{E}_{p_0(s_I,s'_I,s_{\setminus I},x),p_0(v|s_I,s_{\setminus I}),p_0(v'|s'_I,s_{\setminus I})}\|\tilde{q}_{\setminus I}(v,x) - \tilde{q}_{\setminus I}(v',x)\|
\tag{48}
$$

$$
= \mathbb{E}_{p_0(z,z',t,x),p_0(y|z,t),p_0(y'|z',t)}\|\tilde{q}_{\setminus I}(y,t,x) - \tilde{q}_{\setminus I}(y',t,x)\|
\tag{49}
$$

$$
= \mathbb{E}_{p_0(t)}\|\delta_t(\cdot) - \delta_t(\cdot)\|
\tag{50}
$$

$$
= 0.
\tag{51}
$$

Therefore $\tilde{q}$ disentangles $Z$ and $T$. $\qquad\square$

### I.2.2  PROOF OF PROPOSITION 1

**Assumption 3** (Uniqueness). *Two different individuals do not have the exact same outcome (a.s.) under the same treatment $\alpha$, i.e. for $y \sim p(Y|z,\alpha)$, $y' \sim p(Y|z',\alpha)$, we have $y \neq y'$ a.s. if $z \neq z'$.*

The uniqueness assumption is weak and automatically satisfied under common assumptions in prior works such as injective decoder, continuous outcome or diffeomorphism (Khemakhem et al., 2020; von Kügelgen et al., 2023). Note that this assumption could be unrealistic in traditional causal inference but is very reasonable in high-dimensional outcome settings – take facial imaging dataset for example, no two individuals with different facial features should look exactly identical pixel-by-pixel under the same treatment.

*Proof.* Denote

$$
\begin{aligned}
LB(p_., q_.) := \mathbb{E}_{p_0(d)} \big\{ &\mathbb{E}_{q_.(z|y,t,x)} \log \left[ p_.(y|z,t) \right] + \log \left[ \hat{p}(y'|x,t') \right] \\
&- D_{\mathrm{KL}} \left[ q_.(\cdot|y,t,x) \parallel q_.(\cdot|y',t',x) \right] \big\}
\end{aligned}
\tag{52}
$$

and suppose that $(p_*, q_*) \in \arg\max_{p_., q_.} LB(p_., q_.)$ but

$$
\mathbb{E}_{p_0(d)} D_{\mathrm{KL}} \left[ q_*(\cdot|y,t,x) \parallel q_*(\cdot|y',t',x) \right] > 0,
\tag{53}
$$

which we can factorize as

$$
\mathbb{E}_{p_0(x,z,t,t'),p^*} D_{\mathrm{KL}} \left[ q_* \circ p^*(z,t;x) \parallel q_* \circ p^*(z,t';x) \right] > 0
\tag{54}
$$

where $p^*(z,t) = (p_0(Y|z,t), \delta_t(T))$. Then we consider model $(p_{**}, q_{**})$ such that $\forall x, z, t$, we have

$$
q_{**}(Z|f_0(z,t,0),t,x) = \delta_{\gamma(z)}(Z)
\tag{55}
$$

$$
p_{**}(Y|\gamma(z),t) = \delta_{f_0(z,t,0)}(Y)
\tag{56}
$$

where $f_0(z,t,\epsilon_Y) = p_0(Y|z,t)$ is the true SCM and $\gamma$ is any injective transformation of $z$ (e.g. $\gamma(z) = z$). Equation 55 is well-defined by Assumption 2 and 3, which guarantees that $f_0(z,t,0) \neq f_0(z',t,0)$ for $z \neq z'$; Equation 56 is well-defined due to the injectiveness of $\gamma$. $q_{**} = q_*$, $p_{**} = p_*$ elsewhere. It follows that

$$
\begin{aligned}
&\mathbb{E}_{p_0(d)} D_{\mathrm{KL}} \left[ q_{**}(\cdot|y,t,x) \parallel q_{**}(\cdot|y',t',x) \right] \\
&= \mathbb{E}_{p_0(x,z,t,t'),p^*} D_{\mathrm{KL}} \left[ q_* \circ p^*(z,t;x) \parallel q_* \circ p^*(z,t';x) \right]
\end{aligned}
\tag{57}
$$

$$
= \mathbb{E}_{p_0(x,z,t,t')} D_{\mathrm{KL}} \left[ q_{**}(\cdot|f_0(z,t,0),t,x) \parallel q_{**}(\cdot|f_0(z,t',0),t',x) \right]
\tag{58}
$$

$$
= \mathbb{E}_{p_0(x,z,t,t')} D_{\mathrm{KL}} \left[ \delta_{\gamma(z)}(\cdot) \parallel \delta_{\gamma(z)}(\cdot) \right]
\tag{59}
$$

$$
< \mathbb{E}_{p_0(d)} D_{\mathrm{KL}} \left[ q_*(\cdot|y,t,x) \parallel q_*(\cdot|y',t',x) \right]
\tag{60}
$$

by Assumption 2, in the meantime

$$
\begin{aligned}
&\mathbb{E}_{p_0(d)} \mathbb{E}_{q_{**}(z|y,t,x)} \log \left[ p_{**}(y|z,t) \right] \\
&= \mathbb{E}_{p_0(x,z,t),p_0(y|z,t)} \mathbb{E}_{q_{**}(z_{**}|y,t,x)} \log \left[ p_{**}(y|z_{**},t) \right]
\end{aligned}
\tag{61}
$$

$$
= \mathbb{E}_{p_0(x,z,t)} \mathbb{E}_{q_{**}(z_{**}|f_0(z,t,0),x,t)} \log \left[ p_{**}(f_0(z,t,0)|z_{**},t) \right]
\tag{62}
$$

$$
= \mathbb{E}_{p_0(x,z,t)} \log \left[ p_{**}(f_0(z,t,0)|\gamma(z),t) \right]
\tag{63}
$$

$$
= \mathbb{E}_{p_0(x,z,t)} \log \left[ \delta_{f_0(z,t,0)}(f_0(z,t,0)) \right]
\tag{64}
$$

$$
= \mathbb{E}_{p_0(x,z,t)} \mathbb{E}_{q_*(z_*|f_0(z,t,0),x,t)} \log \left[ \delta_{f_0(z,t,0)}(f_0(z,t,0)) \right]
\tag{65}
$$

$$
\geq \mathbb{E}_{p_0(x,z,t)} \mathbb{E}_{q_*(z_*|f_0(z,t,0),x,t)} \log \left[ p_*(y|z_*,t) \right]
\tag{66}
$$

$$
= \mathbb{E}_{p_0(x,z,t),p_0(y|z,t)} \mathbb{E}_{q_*(z_*|y,t,x)} \log \left[ p_*(y|z_*,t) \right]
\tag{67}
$$

$$
= \mathbb{E}_{p_0(d)} \mathbb{E}_{q_*(z|y,t,x)} \log \left[ p_*(y|z,t) \right]
\tag{68}
$$

by Assumption 2. Hence we have

$$
LB(p_{**}, q_{**}) > LB(p_*, q_*)
\tag{69}
$$

which contradicts $(p_*, q_*) \in \arg\max_{p_., q_.} LB(p_., q_.)$. Therefore $(p_*, q_*)$ must satisfy

$$
\mathbb{E}_{p_0(d)} D_{\mathrm{KL}} \left[ q_*(\cdot|y,t,x) \parallel q_*(\cdot|y',t',x) \right] = 0
\tag{70}
$$

and it follows that $\tilde{q}_*$ disentangles $Z$ and $T$ by Lemma 1. $\qquad\square$

### I.2.3 PROOF OF PROPOSITION 2

*Proof.* The strategy is similar to the proof of Proposition 1. We first prove a statement similar to Lemma 1 that states a sufficient condition for empirical disentanglement (**sufficiency**), and then argue that this condition is attainable by optimizing $J_{p_{\mathrm{data}}}(p_., q_.)$ (**attainability**).

**Sufficiency** We prove that if

$$\mathbb{E}_{p_{\text{data}}(x,t,y),p_{\text{data}}(t'|x),p_{\text{data}}(t''|x)}D_{\text{KL}}\left[q(\cdot|y',t',x)\parallel q(\cdot|y'',t'',x)\right]=0 \tag{71}$$

where $y'=\mathbb{E}_{p.(\cdot|\mathbb{E}_{q(\cdot|y,t,x)}Z,t')}Y$ and $y''=\mathbb{E}_{p.(\cdot|\mathbb{E}_{q(\cdot|y,t,x)}Z,t'')}Y$, then $(\tilde{g}.,\tilde{q})$ disentangles $Z$ and $T$ under $p_e=\int_y e\cdot p_{\text{data}}$.

Let $S_I=Z$ and $S_{\backslash I}=T$. On one hand, notice that for any $x$, $z$ s.t. $p_e(x,z)>0$, we have $p_e(t|x,z)>0$ only if $p_e(x,z,t)>0$ only if $p_e(x,t)>0$ only if $p_{\text{data}}(x,t)>0$ only if $p_{\text{data}}(t|x)>0$ for any $t$. It follows that

$$\mathbb{E}_{p_e,\tilde{g}.}\|\tilde{q}_I\circ\tilde{g}.(z,t';x)-\tilde{q}_I\circ\tilde{g}.(z,t'';x)\|$$

$$=\mathbb{E}_{p_{\text{data}}(x,t,y),e(z|x,t,y),p_e(t',t''|x,z),\tilde{g}.}\|\tilde{q}_I\circ\tilde{g}.(z,t';x)-\tilde{q}_I\circ\tilde{g}.(z,t'';x)\| \tag{72}$$

$$=\mathbb{E}_{p_{\text{data}}(x,t,y),p_e(t',t''|x,\mathbb{E}_{q(\cdot|y,t,x)}Z)}\|\tilde{q}_I(y',t',x)-\tilde{q}_I(y'',t'',x)\| \tag{73}$$

$$=\mathbb{E}_{p_{\text{data}}(x,t,y),p_e(t',t''|x,\mathbb{E}_{q(\cdot|y,t,x)}Z)}\|q(\cdot|y',t',x)-q(\cdot|y'',t'',x)\| \tag{74}$$

$$\le\mathbb{E}_{p_{\text{data}},p_e}\sqrt{\frac{1}{2}D_{\text{KL}}\left[q(\cdot|y',t',x)\parallel q(\cdot|y'',t'',x)\right]}\quad\text{(Pinsker's inequality)} \tag{75}$$

$$=0, \tag{76}$$

since Equation 71 implies that $D_{\text{KL}}[q(\cdot|y',t',x)\parallel q(\cdot|y'',t'',x)]=0$ a.e. with respect to $p_{\text{data}}$. Hence $(\tilde{g}.,\tilde{q})$ is consistent with $Z$ under $p_e$. On the other hand, $(\tilde{g}.,\tilde{q})$ is restricted to $Z$ under $p_e$ by design:

$$\mathbb{E}_{p_e,\tilde{g}.}\|\tilde{q}_{\backslash I}\circ\tilde{g}.(z',t;x)-\tilde{q}_{\backslash I}\circ\tilde{g}.(z'',t;x)\|$$

$$=\mathbb{E}_{p_e(x,t),p_e(z',z''|x,t),\tilde{g}.}\|\tilde{q}_{\backslash I}\circ\tilde{g}.(z',t;x)-\tilde{q}_{\backslash I}\circ\tilde{g}.(z'',t;x)\| \tag{77}$$

$$=\mathbb{E}_{p_e(x,t),p_e(z',z''|x,t)}\|\tilde{q}_{\backslash I}(\mathbb{E}_{p.(\cdot|z',t)}Y,t,x)-\tilde{q}_{\backslash I}(\mathbb{E}_{p.(\cdot|z'',t)}Y,t,x)\| \tag{78}$$

$$=\mathbb{E}_p\|\delta_t(\cdot)-\delta_t(\cdot)\| \tag{79}$$

$$=0. \tag{80}$$

Therefore $(\tilde{g}.,\tilde{q})$ disentangles $Z$ and $T$ under $\int_y e\cdot p_{\text{data}}$.

**Attainability** Let $A=\{(x_{k_i},t_{k_i},y_{k_i})\}|_{i=1}^m$ be the set of all unique entries in $\{(x_k,t_k,y_k)\}|_{k=1}^n$. Suppose that $(p_*,q_*)\in\arg\max_{p.,q.}J_{p_{\text{data}}}(p.,q.)$ but

$$\mathbb{E}_{p_{\text{data}}(x,t,y),p_{\text{data}}(t'|x),p_{\text{data}}(t''|x)}D_{\text{KL}}\left[q_*(\cdot|y'_*,t',x)\parallel q_*(\cdot|y''_*,t'',x)\right]>0, \tag{81}$$

where $y'_*=\mathbb{E}_{p_*(\cdot|\mathbb{E}_{q_*(\cdot|y,t,x)}Z,t')}Y$ and $y''_*=\mathbb{E}_{p_*(\cdot|\mathbb{E}_{q_*(\cdot|y,t,x)}Z,t'')}Y$, i.e. $\exists l:1\le l\le n$ and $\exists t^{(1)},t^{(2)}:p_{\text{data}}(t^{(1)}|x_l)\cdot p_{\text{data}}(t^{(2)}|x_l)>0$ such that

$$D_{\text{KL}}\left[q(\cdot|y_*^{(1)},t^{(1)},x_l)\parallel q(\cdot|y_*^{(2)},t^{(2)},x_l)\right]>0 \tag{82}$$

where $y_*^{(1)}=\mathbb{E}_{p_*(\cdot|\mathbb{E}_{q_*(\cdot|y_l,t_l,x_l)}Z,t^{(1)})}Y$ and $y_*^{(2)}=\mathbb{E}_{p_*(\cdot|\mathbb{E}_{q_*(\cdot|y_l,t_l,x_l)}Z,t^{(2)})}Y$. Then we consider model $(p_{**},q_{**})$ such that $\forall(x,t,y)\in A$, we have

$$q_{**}(Z|\mathbb{E}_{p_*(\cdot|\mathbb{E}_{q_*(\cdot|y,t,x)}Z,\alpha)}Y,\alpha,x)=q_*(Z|y,t,x) \tag{83}$$

for any $\alpha$ s.t. $p_{\text{data}}(T=\alpha|x)>0$. Equation 83 is well-defined since

$$\mathbb{E}_{p_*(\cdot|\mathbb{E}_{q_*(\cdot|y,t,x)}Z,\alpha)}Y=\mathbb{E}_{p_*(\cdot|\mathbb{E}_{q_*(\cdot|y',t',x)}Z,\alpha)}Y$$

$$\Leftrightarrow g_*(Y|\mathbb{E}_{q_*(\cdot|y,t,x)}Z,\alpha)=g_*(Y|\mathbb{E}_{q_*(\cdot|y',t',x)}Z,\alpha)\Rightarrow\mathbb{E}_{q_*(\cdot|y,t,x)}Z=\mathbb{E}_{q_*(\cdot|y',t',x)}Z \tag{84}$$

$$\Leftrightarrow e_*(Z|y,t,x)=e_*(Z|y',t',x)\Rightarrow(y,t)=(y',t')\Rightarrow q_*(Z|y,t,x)=q_*(Z|y',t',x) \tag{85}$$

by uniqueness condition 1) and the construction of $\{q.\}$. $q_{**}=q_*$ elsewhere and $p_{**}=p_*$ everywhere.

Firstly, we show that $(p_{**},q_{**})$ is still a member of the model class of interest. Since $q_{**}$ is still normally distributed with unit variance by definition and $g_{**}$ still satisfies uniqueness condition 1) by definition, we only need to show that $(g_{**},e_{**})$ still satisfies uniqueness condition 2). This

is because $\forall x, t, y, t' : p_{\text{data}}(x, t, y) \cdot p_{\text{data}}(t'|x) > 0$, we have $(x, t, y) \in A$ and thus $(x, t, y) = (x, \alpha, \mathbb{E}_{p_*(\cdot|\mathbb{E}_{q_*(\cdot|y_{k_i}, t_{k_i}, x_{k_i})}Z, \alpha)}Y)$ for some $(x_{k_i}, t_{k_i}, y_{k_i}) \in A$ only if

$$(x, \alpha, \mathbb{E}_{p_*(\cdot|\mathbb{E}_{q_*(\cdot|y_{k_i}, t_{k_i}, x_{k_i})}Z, \alpha)}Y) = (x, \alpha, \mathbb{E}_{g_*(\cdot|z, t')e_*(z|y_{k_i}, t_{k_i}, x_{k_i})}Y) \in A \tag{86}$$

only if $(x, \alpha, \mathbb{E}_{p_*(\cdot|\mathbb{E}_{q_*(\cdot|y_{k_i}, t_{k_i}, x_{k_i})}Z, \alpha)}Y) = (x_{k_i}, t_{k_i}, y_{k_i})$ by the fact that $(g_*, e_*)$ satisfies uniqueness condition 2). In this case, we have $(x, t, y) = (x_{k_i}, t_{k_i}, y_{k_i})$ and

$$q_{**}(Z|y, t, x) = q_{**}(Z|\mathbb{E}_{p_*(\cdot|\mathbb{E}_{q_*(\cdot|y_{k_i}, t_{k_i}, x_{k_i})}Z, \alpha)}Y, \alpha, x) \tag{87}$$

$$= q_*(Z|y_{k_i}, t_{k_i}, x_{k_i}) = q_*(Z|y, t, x). \tag{88}$$

If $(x, t, y) \neq (x, \alpha, \mathbb{E}_{p_*(\cdot|\mathbb{E}_{q_*(\cdot|y_{k_i}, t_{k_i}, x_{k_i})}Z, \alpha)}Y)$ for any $(x_{k_i}, t_{k_i}, y_{k_i}) \in A$, then $q_{**}(Z|y, t, x) = q_*(Z|y, t, x)$ by definition. Therefore we have $q_{**}(Z|y, t, x) = q_*(Z|y, t, x)$. It follows that $y'_{**} = \mathbb{E}_{g_{**}(\cdot|z, t')e_{**}(z|y, t, x)}Y = \mathbb{E}_{g_*(\cdot|z, t')e_*(z|y, t, x)}Y = y'_*$ and $(g_{**}, e_{**})$ satisfies uniqueness condition 2) by the fact that $(g_*, e_*)$ satisfies uniqueness condition 2).

Secondly, we show that $(p_*, q_*)$ is no longer optimal, thus raise a contradiction. On one hand, we must have

$$\mathbb{E}_{p_{\text{data}}} D_{\text{KL}}\left[ q_*(\cdot|y, t, x) \,\|\, q_*(\cdot|y'_*, t', x) \right]$$

$$\geq p_{\text{data}}(x_l, t_l, y_l) \left\{ p_{\text{data}}(t^{(1)}|x_l) D_{\text{KL}}\left[ q_*(\cdot|y_l, t_l, x_l) \,\|\, q_*(\cdot|y_*^{(1)}, t^{(1)}, x_l) \right] \right.$$

$$\left. + p_{\text{data}}(t^{(2)}|x_l) D_{\text{KL}}\left[ q_*(\cdot|y_l, t_l, x_l) \,\|\, q_*(\cdot|y_*^{(2)}, t^{(2)}, x_l) \right] \right\} \tag{89}$$

$$> 0, \tag{90}$$

otherwise we would have $q_*(Z|y_l, t_l, x_l) = q_*(Z|y_*^{(1)}, t^{(1)}, x_l) = q_*(Z|y_*^{(2)}, t^{(2)}, x_l)$ a.e. which contradicts Equation 82. Hence

$$\mathbb{E}_{p_{\text{data}}} D_{\text{KL}}\left[ q_{**}(\cdot|y, t, x) \,\|\, q_{**}(\cdot|y'_{**}, t', x) \right]$$

$$= \sum_{(x, t, y) \in A} p_{\text{data}}(x, t, y) \sum_{t'} p_{\text{data}}(t'|x) \cdot D_{\text{KL}}\left[ q_*(\cdot|y, t, x) \,\|\, q_{**}(\cdot|y'_{**}, t', x) \right] \tag{91}$$

$$= \sum_{(x, t, y) \in A} p_{\text{data}}(x, t, y) \sum_{t'} p_{\text{data}}(t'|x) \cdot D_{\text{KL}}\left[ q_*(\cdot|y, t, x) \,\|\, q_*(\cdot|y, t, x) \right] \tag{92}$$

$$= 0 < \mathbb{E}_{p_{\text{data}}} D_{\text{KL}}\left[ q_*(\cdot|y, t, x) \,\|\, q_*(\cdot|y'_*, t', x) \right] \tag{93}$$

by Equation 83 and the fact that $q_{**}(Z|y, t, x) = q_*(Z|y, t, x)$ for any $(x, t, y) \in A$ as shown above. On the other hand, we have

$$\mathbb{E}_{p_{\text{data}}} \left\{ \mathbb{E}_{q_{**}(z|y, t, x)} \log\left[ p_{**}(y|z, t) \right] + \log\left[ \hat{p}(y'_{**}|x, t') \right] \right\}$$

$$= \mathbb{E}_{p_{\text{data}}} \left\{ \mathbb{E}_{q_*(z|y, t, x)} \log\left[ p_*(y|z, t) \right] + \log\left[ \hat{p}(y'_*|x, t') \right] \right\} \tag{94}$$

simply by the construction of $p_{**}$ and the fact that $q_{**}(Z|y, t, x) = q_*(Z|y, t, x)$, $y'_{**} = y'_*$ for any $(x, t, y) \in A$ as shown above. Hence we have

$$J_{p_{\text{data}}}(p_{**}, q_{**}) > J_{p_{\text{data}}}(p_*, q_*) \tag{95}$$

which contradicts $(p_*, q_*) \in \arg\max_{p., q.} J_{p_{\text{data}}}(p., q.)$. Therefore $(p_*, q_*)$ must satisfy

$$\mathbb{E}_{p_{\text{data}}(x, t, y), p_{\text{data}}(t'|x), p_{\text{data}}(t''|x)} D_{\text{KL}}\left[ q_*(\cdot|y'_*, t', x) \,\|\, q_*(\cdot|y''_*, t'', x) \right] = 0 \tag{96}$$

and it follows that $(\tilde{g}_*, \tilde{q}_*)$ disentangles $Z$ and $T$ under $\int_y e_* p_{\text{data}}$ by **sufficiency**. $\qquad\square$

### I.2.4 PROOF OF COROLLARY 1

*Proof.* By the proof of Proposition 2 (attainability), we have

$$\mathbb{E}_{p_{\text{data}}(x, t, y), p_{\text{data}}(t'|x), p_{\text{data}}(t''|x)} D_{\text{KL}}\left[ q_*(\cdot|y'_*, t', x) \,\|\, q_*(\cdot|y''_*, t'', x) \right] = 0 \tag{97}$$

where $y'_* = \mathbb{E}_{p_*(\cdot|\mathbb{E}_{q_*(\cdot|y, t, x)}Z, t')}Y$ and $y''_* = \mathbb{E}_{p_*(\cdot|\mathbb{E}_{q_*(\cdot|y, t, x)}Z, t'')}Y$, hence $\mathbb{E}_{q_*(\cdot|y'_*, t', x)}Z = \mathbb{E}_{q_*(\cdot|y''_*, t'', x)}Z$ a.e. on $p_{\text{data}}$ and it follows that

$$\mathbb{E}_{p_{\text{data}}(x, t, y), p_{\text{data}}(t'|x), p_{\text{data}}(t''|x)} D_{\text{KL}}\left[ e_*(\cdot|y'_*, t', x) \,\|\, e_*(\cdot|y''_*, t'', x) \right]$$

$$= \mathbb{E}_{p_{\text{data}}(x, t, y), p_{\text{data}}(t'|x), p_{\text{data}}(t''|x)} D_{\text{KL}}\left[ \delta_{\mathbb{E}_{q_*(\cdot|y'_*, t', x)}Z}(\cdot) \,\|\, \delta_{\mathbb{E}_{q_*(\cdot|y''_*, t'', x)}Z}(\cdot) \right] = 0. \tag{98}$$

Notice that $y'_* = \mathbb{E}_{p_*(\cdot|\mathbb{E}_{e_*(\cdot|y, t, x)}Z, t')}Y$ and $y''_* = \mathbb{E}_{p_*(\cdot|\mathbb{E}_{e_*(\cdot|y, t, x)}Z, t'')}Y$, we have $(\tilde{g}_*, \tilde{e}_*)$ disentangles $Z$ and $T$ under $\int_y e_* p_{\text{data}}$ by the proof of Proposition 2 (sufficiency). $\qquad\square$

### I.3 MARGINAL ESTIMATOR

### I.3.1 PROOF OF THEOREM 2

*Proof.* $\Psi(p)$ has the identification $\Psi(p) = \mathbb{E}_p[\mathbb{E}_p[Y'|Z, X, T' = \alpha]] = \mathbb{E}_p[\mathbb{E}_p[Y'|Z, T' = \alpha]]$ under Figure 1. Following Van der Vaart (2000), we define a path $p_\epsilon(\Lambda) = p(\Lambda)(1 + \epsilon S(\Lambda))$ on density $p$ of random vector $\Lambda$ as a submodel that passes through $p$ at $\epsilon = 0$ in the direction of the score $S(\Lambda) = \frac{d}{d\epsilon} \log[p_\epsilon(\Lambda)]\Big|_{\epsilon=0}$. Following the key identity presented in Levy (2019):

$$\frac{d}{d\epsilon} p_\epsilon(\lambda_i|pa(\Lambda_i) = \bar{\lambda}_{i-1})\Big|_{\epsilon=0}$$
$$= p(\lambda_i|pa(\Lambda_i) = \bar{\lambda}_{i-1})(\mathbb{E}[S(\Lambda)|\Lambda_i = \lambda_i, pa(\Lambda_i) = \bar{\lambda}_{i-1}] - \mathbb{E}[S(\Lambda)|pa(\Lambda_i) = \bar{\lambda}_{i-1}]) \quad (99)$$

where $pa(\Lambda_i)$ denotes the parent nodes of variable $\Lambda_i$, we have

$$\frac{d}{d\epsilon}\Psi(p_\epsilon)\Big|_{\epsilon=0} = \frac{d}{d\epsilon}\Big|_{\epsilon=0} \mathbb{E}_{p_\epsilon}\left[\mathbb{E}_{p_\epsilon}\left[Y'|Z, T' = \alpha\right]\right] \quad (100)$$

$$= \frac{d}{d\epsilon}\Big|_{\epsilon=0} \int_{y',z,x} y' \left[p_\epsilon(y'|z, T' = \alpha)p_\epsilon(z, x)\right] \quad (101)$$

$$= \int_{y',z,x} y' \frac{d}{d\epsilon}\Big|_{\epsilon=0} \left[p_\epsilon(y'|z, T' = \alpha)p_\epsilon(z, x)\right] \quad \text{(dominated convergence)} \quad (102)$$

$$= \int_{y',z,x} y' p(z, x) \frac{d}{d\epsilon}\Big|_{\epsilon=0} p_\epsilon(y'|z, T' = \alpha) \quad (103)$$

$$+ \int_{y',z,x} y' p(y'|z, T' = \alpha) \frac{d}{d\epsilon}\Big|_{\epsilon=0} p_\epsilon(z, x) \quad (104)$$

$$= \int_w I(t' = \alpha) \frac{p(t'|x)}{p(t'|x)} y' p(y, t|z, x) p(z, x) \frac{d}{d\epsilon}\Big|_{\epsilon=0} p_\epsilon(y'|z, t')$$
$$+ \int_{y',z,x} y' p(y'|z, T' = \alpha) \frac{d}{d\epsilon}\Big|_{\epsilon=0} p_\epsilon(z, x) \quad (105)$$

$$= \int_w \frac{I(t' = a)}{p(t'|x)} y' p(y', t'|z, x) p(y, t|z, x) p(z, x) \left\{S(w) - \mathbb{E}\left[S(W)|y, z, x, t, t'\right]\right\}$$
$$+ \int_{y',z,x} y' p(y'|z, T' = \alpha) p(z, x) \left\{\mathbb{E}\left[S(W)|z, x\right] - \mathbb{E}\left[S(W)\right]\right\} \quad (106)$$

$$= \int_w \frac{I(t = a)}{p(t|x)} y p(y, t|z, x) p(y', t'|z, x) p(z, x) \left\{S(w) - \mathbb{E}\left[S(W)|y', z, x, t', t\right]\right\}$$
$$+ \int_{y',z,x} y' p(y'|z, T' = \alpha) p(z, x) \left\{\mathbb{E}\left[S(W)|z, x\right] - \mathbb{E}\left[S(W)\right]\right\} \quad (107)$$

$$= \int_w S(w) \cdot \frac{I(t = \alpha)}{p(t|x)} y p(w)$$
$$- \int_w \mathbb{E}\left[S(W)|y', z, x, t, t'\right] p(y', z, x, t, t') \cdot \frac{I(t = \alpha)}{p(t|x)} y p(y|z, t)$$
$$+ \int_{y',z,x} \mathbb{E}\left[S(W)|z, x\right] p(z, x) \cdot y' p(y'|z, T' = \alpha)$$
$$- \int_{y',z,x} \mathbb{E}\left[S(W)\right] \cdot y' p(y'|z, T' = \alpha) p(z, x) \quad (108)$$

$$= \int_w S(w) \left\{ \frac{I(t = \alpha)}{p(t|x)}(y - \mathbb{E}\left[Y|z, t\right]) + \mathbb{E}\left[Y'|z, T' = \alpha\right] - \Psi \right\} p(w) \quad (109)$$

by the assumption of Theorem 2 and factorization according to Figure 1. Hence

$$\frac{d}{d\epsilon}\Psi(p_\epsilon)\Big|_{\epsilon=0} = \left\langle S(W), \frac{I(T = \alpha)}{p(T|X)}(Y - \mathbb{E}_p\left[Y|Z, T\right]) + \mathbb{E}_p\left[Y'|Z, T' = \alpha\right] - \Psi \right\rangle_{L^2(\Omega;E)} \quad (110)$$

and we have $\tilde{\psi}_p = I(T = \alpha)/p(T|X) \cdot (Y - \mathbb{E}_p[Y|Z, T]) + \mathbb{E}_p[Y'|Z, T' = \alpha] - \Psi$. $\quad\square$

### I.3.2 PROOF OF COROLLARY 3

*Proof.* The proof is a simple extension on the proof of Theorem 2. We have

$$\frac{d}{d\epsilon}\Xi(p_\epsilon)\Big|_{\epsilon=0} = \frac{d}{d\epsilon}\Big|_{\epsilon=0}\mathbb{E}_{p_\epsilon}\left[\mathbb{E}_{p_\epsilon}\left[Y'|Z,T'=\alpha\right]|X=c\right] \tag{111}$$

$$= \frac{d}{d\epsilon}\Big|_{\epsilon=0}\int_{y',z,x} y'\left[p_\epsilon(y'|z,T'=\alpha)p_\epsilon(z|X=c)\right]. \tag{112}$$

Hence, following the same derivation, $p_\epsilon(z,x)$ in Equation 102 and onwards becomes $p_\epsilon(z|X=c)$, while $p(z,x)$ in Equation 105 and onwards becomes $I(x=c)/p(x) \cdot p(z,x)$. Therefore, we have $\tilde{\xi}_p(W) = I(X=c)/p(X) \cdot \tilde{\psi}_p(W)$. □

## J MODEL ARCHITECTURE FOR IMAGE GENERATION TASKS

The specific model architecture used for image generation can be found in Figure 8. Note that in the image generation tasks in our experiments, we treated every condition as intervenable, thus every condition belongs in $T$ rather than $X$. If this is not the case, the non-intervenable conditions would compose covariates $X$ and would only be inputted to the encoding model but not the decoding model.

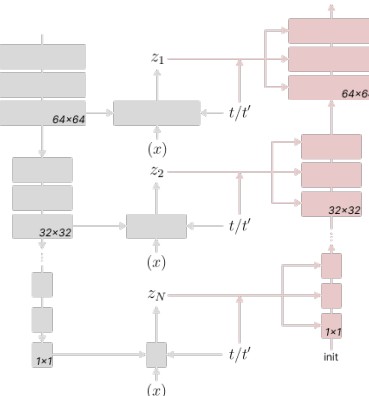

Figure 8: Model architecture of the latent recognition model (grey) and outcome construction model (pink) for image generation tasks.

To incorporate $t$ $(t')$ and $x$ into the convolutional layers, we first embed them into vector representations (tabular embedding for categorical variables; sinusoidal embedding with non-linear mapping for continuous variable), then repeat and expand each value of the vectors to match the input resolutions of the corresponding blocks and concatenate them to image representations as extra channels, similar to Monteiro et al. (2023). Note that there are other ways to conduct image-vector incorporation such as the scale-shift approach commonly used in diffusion models to incorporate time step variable (Sanchez & Tsaftaris, 2022). Each convolutional block (each rectangle in Figure 8) contains a $1\times1$ embedding layer, a $3\times3$ (or $1\times1$ if resolution is lower than $3\times3$) botteleneck layer, and a $3\times3$ (or $1\times1$ if resolution is lower than $3\times3$) output layer.

## K DETACHING PATTERNS

In practice, there are a few optional gradient detaching options when evaluating the divergence term to enhance the stability of stochastic optimization. Our default behavior is to use a copy of $q_\phi$ for the evaluation of the divergence term which is updated after every epoch, while preserving the gradient path and gradient computation for $y'_{\theta,\phi}$. This technique is analogous to the handling of surrogate objectives commonly applied in reinforcement learning (Lillicrap et al., 2015; Van Hasselt et al., 2016; Schulman et al., 2017), and the copy of $q_\phi$ is seen as the target network. In addition, we provide the option of detaching $\phi$ from $y'_{\theta,\phi}$ (focusing on the second purpose described in divergence interpretation: preservation of individuality in decoding) and the option of not detaching the gradient

computation for $q_\phi$ (reflecting the first purpose in divergence interpretation: recognition of mutual features in encoding) during the evaluation of the latent divergence term.

## L    BENCHMARK ADAPTATIONS

### L.1    AUTOENCODER

The adapted autoencoder reconstructs the outcome during training similar to a generic autoencoder, but takes treatment and covariates as additional inputs. During test time, we simply plug in the counterfactual treatments along with factual outcomes and covariates to generate the counterfactual outcome predictions.

### L.2    GANITE

GANITE (Yoon et al., 2018)'s counterfactual generator does not scale with a combination of high-dimensional outcome and multi-level treatment, thus here we only input one randomly sampled counterfactual treatment to the generator and correspondingly construct one counterfactual outcome for each sample. See Figure 9 for the original and adapted structure of the model.

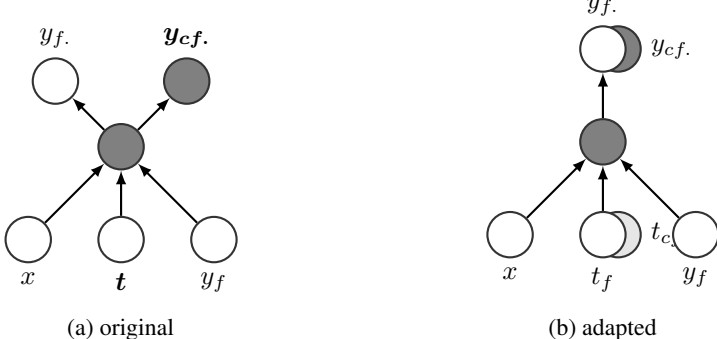

(a) original                        (b) adapted

Figure 9: GANITE's counterfactual generator. $t_{cf}$ is a random sample of $t$, passed into the generator as a part of the input $(x, t_{cf}, y_f)$, separately from input $(x, t_f, y_f)$ of the factual generation.

The discriminator predicts the logits $l_f$, $l_{cf}$ of $y_f$, $y_{cf.}$ separately. The cross entropy loss of $(l_f, l_{cf})$ against $(1, 0)$ is then calculated.

## M    EXPERIMENT SETTINGS

In this section, we describe the settings of our experiments in detail. For complete information on hyperparameter settings, see our codebase at `https://github.com/yulun-rayn/variational-causal-inference`.

### M.1    SINGLE-CELL PERTURBATION DATASETS

For both datasets in our experiments, two thousand most variable genes were selected for training and testing. Marson has 2,000 dimensional outcomes, 3 categorical covariates (cell type, donor indicator, stimulation indicator) and 1 categorical treatment (target gene), Sciplex has 2,000 dimensional outcomes, 2 categorical covariates (cell type, replicate indicator) and 1 categorical treatment (used drug). Data with certain treatment-covariate combinations are held out as the out-of-distribution (OOD) set and the rest are split into training and validation set with a four-to-one ratio.

**Out-of-Distribution Selections**   We randomly select a covariate category (e.g. a cell type) and hold out all cells in this category that received one of the twenty perturbations whose effects are the hardest to predict. We use these held-out data to compose the out-of-distribution (OOD) set. We computed the Euclidean distance between the pseudobulked gene expression of each perturbation

against the rest of the dataset, and selected the twenty most distant ones as the hardest-to-predict perturbations. This is the same procedure carried out by Lotfollahi et al. (2021).

**Differentially-Expressed Genes**  In order to evaluate the quality of the predictions on the genes that were substantially affected by the perturbations, we select sets of 50 differentially-expressed (DE) genes associated with each perturbation and separately evaluate model performance on these genes. This is the same procedure carried out by Lotfollahi et al. (2021).

**Hyperparameter Settings**  All common hyperparameters of all models are set to the same as the defaults of CPA (Lotfollahi et al., 2021): an universal number of hidden dimensions $128$; number of layers 6 (encoder 3, decoder 3); an universal learning rate $3^{-4}$, weight decay rate $4^{-7}$. Contrary to CPA, we use step-based learning rate decay instead of epoch-based learning rate decay, and decay step size is set to $400,000$ while decay rate remains the same at $0.1$. Batch size is $64$ for Marson and $128$ for Sciplex.

**Other Details**  We used the empirical outcome distribution (with Gaussian kernel smoother) stratified by $X$ and $T$ to estimate the covariate-specific outcome model $p(Y|X,T)$. The fully-attached detaching pattern is applied where both $y'_{\theta,\phi}$ and $q_\phi$ are not detached when evaluating the divergence term. Models are trained on Amazon web services' accelerated computing EC2 instance G4dn which contains 2nd Generation Intel Xeon Scalable Processors (Cascade Lake P-8259CL) and up to 8 NVIDIA T4 Tensor Core GPUs.

## M.2 MORPHO-MNIST

We used the original Morpho-MNIST training set for model training, and the original Morpho-MNIST testing set as the observed samples for model testing. The training and testing set have $28{\times}28$ dimensional outcomes, no covariates and 2 continuous treatments (thickness, intensity). Three different testing sets are counstructed based on the counterfactual treatment settings: only modifying thickness for $do(\text{th})$; only modifying intensity for $do(\text{in})$; randomly modifying thickness or intensity for mix. For each observed sample in the testing set, we randomly sample the amount of modification based on the range of the selected modification.

**Data Augmentation**  For each image in the training set ($28{\times}28$), we pad it by a margin of 4 pixels on each side ($36{\times}36$), then randomly crop a $32{\times}32$ region. This is the same procedure carried out by Ribeiro et al. (2023).

**Hyperparameter Settings**  The model width, depth and resolutions of our framework are set to the same as Ribeiro et al. (2023): number of channels for the hierarchical encoding blocks are set to $\{16, 32, 64, 128, 256\}$ with resolutions $\{32, 16, 8, 4, 1\}$ and number of blocks $\{4, 4, 4, 4, 4\}$; number of channels for the hierarchical decoding blocks are set to $\{256, 128, 64, 32, 16\}$ with resolutions $\{1, 4, 8, 16, 32\}$ and number of blocks $\{4, 4, 4, 4, 4\}$. Each block contains a bottleneck layer with one-forth the number of channels as the input/output number of channels. Training is conducted with a batch size of 32, an universal learning rate $1^{-4}$, and weight decay rate $4^{-5}$. Learning rate decays at epoch 100 and linearly decays to 0 at epoch 200.

**Other Details**  We used the adversarial training approach to estimate the covariate-specific outcome model $p(Y|X,T)$. The default detaching pattern (see Appendix K) is applied where a copy of $q_\phi$ is used as critic to evaluate the divergence term, and updated after every epoch. Models are trained on Amazon web services' accelerated computing EC2 instance G4dn which contains 2nd Generation Intel Xeon Scalable Processors (Cascade Lake P-8259CL) and up to 8 NVIDIA T4 Tensor Core GPUs.

## M.3 CELEBA-HQ

The two factors of interest – smiling and glasses, are regarded as intervenable. Therefore, the training and testing set have $64{\times}64{\times}3$ dimensional outcomes, no covariates and 2 categorical treatments. The train-test split of the CelebA-HQ dataset is inherited from the original CelebA dataset.

**Data Augmentation** For each image, we crop the $128 \times 128$ center region, then randomly crop a $120 \times 120$ region within the center region. Afterwards, we resize this region to $64 \times 64$, and apply horizontal flip with probability $0.5$. Note that data augmentation is applied to both training and testing set in this experiment, since we are sampling and empirically examining the counterfactual results and not evaluating any metric.

**Hyperparameter Settings** The model width and resolutions of our framework are set to the same as Monteiro et al. (2023): number of channels for the hierarchical encoding blocks are set to $\{32, 64, 128, 256, 512, 1024\}$ with resolutions $\{64, 32, 16, 8, 4, 1\}$; number of channels for the hierarchical decoding blocks are set to $\{1024, 512, 256, 128, 64, 32\}$ with resolutions $\{1, 4, 8, 16, 32, 64\}$. The number of blocks is set to $\{3, 12, 12, 6, 3, 3\}$ for the encoder and $\{3, 3, 6, 12, 12, 3\}$ for the decoder. This is chosen to be slightly more shallow than Monteiro et al. (2023) ($\{4, 12, 12, 8, 4, 4\}$ and $\{4, 4, 8, 12, 12, 4\}$). Each block contains a bottleneck layer with one-forth the number of channels as the input/output number of channels. Training is conducted with a batch size of 32. An universal learning rate $3^{-4}$, and weight decay rate $4^{-7}$ are inherited from the experiments with single-cell perturbation datasets. Step size for learning rate decay is $400,000$ with decay rate $0.1$.

**Other Details** We used the adversarial training approach to estimate the covariate-specific outcome model $p(Y|X, T)$. The default detaching pattern (see Appendix K) is applied where a copy of $q_\phi$ is used as critic to evaluate the divergence term, and updated after every epoch. Models are trained on Amazon web services' accelerated computing EC2 instance P3 which contains high frequency Intel Xeon Scalable Processor (Broadwell E5-2686 v4) and up to 8 NVIDIA Tesla V100 GPUs, each pairing 5,120 CUDA Cores and 640 Tensor Cores.

# N ADDITIONAL EXPERIMENTS AND DETAILS

## N.1 MARGINAL ESTIMATIONS ON SINGLE-CELL PERTURBATION DATASETS

In this section, we use the same evaluation metric as Section 3.1, but compute each $R^2$ with the robust marginal estimator and compare the results to that of the empirical mean estimator. Note that on OOD set, the robust estimator reduces to empirical mean since no perturbation-covariates combination exist in validation set. Therefore, we compute the $R^2$ of the marginal estimators using samples from the training set against the true empirical average on the validation set in these experiments. In each run, we train a VCI model for individualized outcome predictions, and calculate the evaluation metric for each marginal estimator periodically during the course of training. Table 5 shows the results on Marson. Note that the goal of robust estimation is to produce less biased estimators with tighter confidence bounds, hence we report the mean and standard error over five independent runs to reflect its effectiveness regarding this goal.

Table 5: Comparison of marginal estimators on Marson (Schmidt et al., 2022)

| | All Genes | | DE Genes | |
| --- | --- | --- | --- | --- |
| Episode | mean | robust | mean | robust |
| 40 | $0.9141 \pm 0.0159$ | $\mathbf{0.9343 \pm 0.0080}$ | $0.7108 \pm 0.0735$ | $\mathbf{0.9146 \pm 0.0305}$ |
| 80 | $0.9171 \pm 0.0104$ | $\mathbf{0.9349 \pm 0.0068}$ | $0.7274 \pm 0.0462$ | $\mathbf{0.9163 \pm 0.0254}$ |
| 120 | $0.9204 \pm 0.0097$ | $\mathbf{0.9352 \pm 0.0063}$ | $0.7526 \pm 0.0447$ | $\mathbf{0.9182 \pm 0.0229}$ |
| 160 | $0.9157 \pm \mathbf{0.0043}$ | $\mathbf{0.9355} \pm 0.0053$ | $0.7383 \pm 0.0402$ | $\mathbf{0.9203 \pm 0.0199}$ |

As is shown in the table, the robust estimator provides a crucial adjustment to the empirical mean of model predictions especially on the hard-to-predict elements of high-dimensional vectors. Such estimation could be valuable in many contexts involving high-dimensional predictions where deep learning models might plateau at rather low ceilings.

## N.2 MORE COMPLETE ABLATION STUDY

Note that the latent divergence term is essentially a regularization term to counterfactual supervision, as discussed in Section 2.2, hence it would not be meaningful to train VCI with the latent divergence

term and without the counterfactual supervision term – it would just further worsen the training-inference disconnection problem described in Section 1 without any real benefit. However, we acknowledge that presenting this case would make the ablation study more complete, and therefore have made this inclusion (marked as HAE-A) in Figure 10. As expected, HAE-A is even worse than HAE.

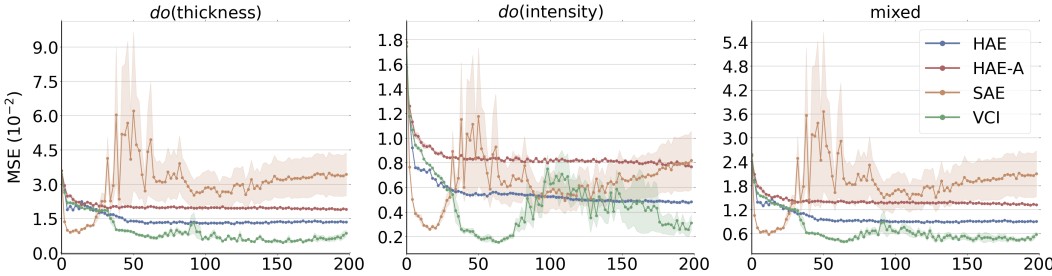

Figure 10: Ablation Study: Added HAE-A representing training VCI with the latent divergence term and without the counterfactual supervision term.

### N.3 EVALUATION ON LATENT DISENTANGLEMENT

The definition of disentanglement (Definition 3) is extended from a well-established definition of disentanglement in Shu et al. (2019), based on the oracle consistency (Definition 1) of the latent representation. It is a different notion from latent disentanglement in the non-linear ICA literatures (Appendix A) although sometimes sharing a non-distinguishable terminology. In this context, the best way to evaluate disentanglement is by definition, using oracle consistency, and we present the ablation results in Table 6. Note that the oracle consistency is measured by KL-divergence for computational efficiency due to the lack of analytic form for the total variational distance between Gaussians. As stated by the Pinsker's Inequality, the total variational distance is bounded once KL-divergence is controlled.

Table 6: Oracle consistency under different strengths (denoted as coefficient $\omega_2$) on the exogenous disentanglement term.

| $\omega_2$ | oracle consistency $\downarrow$ |
|---|---|
| 0 | $3979.86 \pm 202.42$ |
| 0.01 | $0.84 \pm 0.13$ |
| 0.02 | $0.75 \pm 0.11$ |
| 0.03 | $0.53 \pm 0.10$ |
| 0.04 | $0.51 \pm 0.07$ |
| 0.05 | $0.51 \pm 0.08$ |

### N.4 COMPOSITION, EFFECTIVENESS, AND REVERSIBILITY

We examine our model through the composition, effectiveness, and reversibility metrics (Monteiro et al., 2023) and use the best benchmark model in Table 2 as a reference. The results are shown in Table 7. Note that these results only serve model inspection purposes and should not be used as strict performance evaluations in favor of Table 2. These metrics are merely surrogate measurements of counterfactual models' axiomatic characteristics, and they are inconclusive in deciding the best-performing counterfactual model. For example, a naive model that simply returns the original image as counterfactual outcome at all times is completely useless for counterfactual inference, but will achieve the best score on 2 of these 3 metrics – a perfect composition score of 0 and a perfect reversibility score of 0. There is a trade-off between composition and effectiveness as well as effectiveness and reversibility, and it is not conclusive which counterfactual model performs the best unless there is a clear domination of one model over the others on all three metrics.

Table 7: Composition, effectiveness, and reversibility of ours against the best benchmark model in Table 2. Images are scaled to $[0, 1]$ in evaluations. All losses are on the L-2 scale (note that the factors measured by effectiveness – thickness and intensity – are continuous values).

|  | $\beta$ | composition ↓ | effectiveness ↓ | reversibility ↓ |
|---|---|---|---|---|
| MED | 1 | $0 \pm 0$ | $283.99 \pm 67.76$ | $0.01024 \pm 0.00767$ |
| MED | 3 | $0 \pm 0$ | $151.13 \pm 86.46$ | $0.00816 \pm 0.00306$ |
| SAE |  | $0.00048 \pm 0.000084$ | $189.03 \pm 22.92$ | $0.00102 \pm 0.00019$ |
| VCI |  | $0.00026 \pm 0.000055$ | $17.64 \pm 6.23$ | $0.00076 \pm 0.00015$ |

As discussed in Section 3.3, it is expected that our model does not reach a composition of 0 because the consistency assumption does not necessarily hold strictly and the model does not necessarily perform exact reconstruction in regions directly affected by the intervention, which is not an issue for the purpose of counterfactual generative modeling. Note that the composition metric does not interact with counterfactual interventions at all, while the other two metrics interact with counterfactual interventions at least in some capacity.

## N.5 More Details on Morpho-MNIST Experiments

Table 8 shows the standard error of metrics reported in Table 2 across five independent runs. Figure 11 shows the violin plot of errors reported in Table 2 and Table 8 across the corresponding runs. Figure 12 and Figure 13 show some sampled results from VCI.

Table 8: Standard error of metrics.

|  | $\beta$ | Std. of Image MSE ($\cdot 10^{-2}$) | | | Std. of Thickness (th) MAE | | | Intensity (in) MAE ($\cdot 10^{-1}$) | | |
|---|---|---|---|---|---|---|---|---|---|---|
|  |  | $do$(th) | $do$(in) | mix | $do$(th) | $do$(in) | mix | $do$(th) | $do$(in) | mix |
| DEAR |  | 0.20 | 0.06 | 0.11 | 0.18 | 0.10 | 0.21 | 0.32 | 0.49 | 0.25 |
| Diff-SCM |  | 0.02 | 0.02 | 0.02 | 0.01 | 0.01 | 0.01 | 0.01 | 0.02 | 0.01 |
| CHVAE | 1 | 0.53 | 0.86 | 0.62 | 0.03 | 0.15 | 0.10 | 0.33 | 0.32 | 0.20 |
| CHVAE | 3 | 0.90 | 0.83 | 0.94 | 0.04 | 0.21 | 0.08 | 0.19 | 0.10 | 0.17 |
| MED | 1 | 0.92 | 0.26 | 0.59 | 0.11 | 0.08 | 0.12 | 0.31 | 0.19 | 0.23 |
| MED | 3 | 1.34 | 0.40 | 0.86 | 0.05 | 0.06 | 0.07 | 0.25 | 0.09 | 0.15 |
| SAE |  | 0.12 | 0.05 | 0.05 | 0.10 | 0.04 | 0.05 | 0.15 | 0.10 | 0.08 |
| VCI |  | 0.07 | 0.01 | 0.07 | 0.03 | 0.01 | 0.05 | 0.11 | 0.05 | 0.06 |

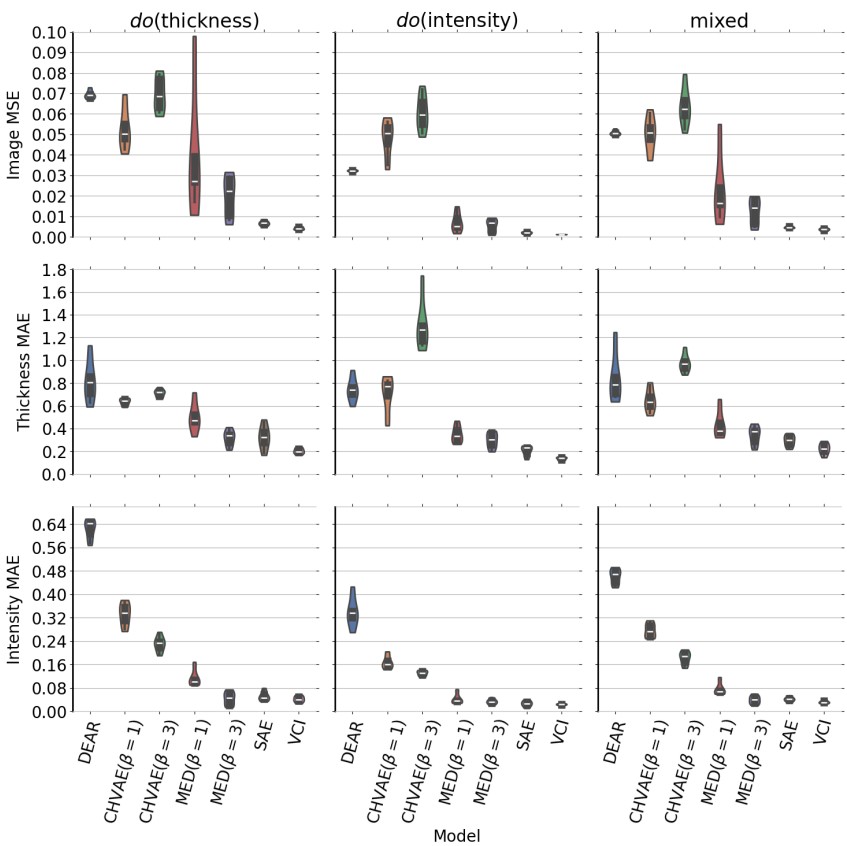

Figure 11: Violin plot of Image MSE, Thickness MAE and Intensity MAE on benchmark models and ours across five independent runs.

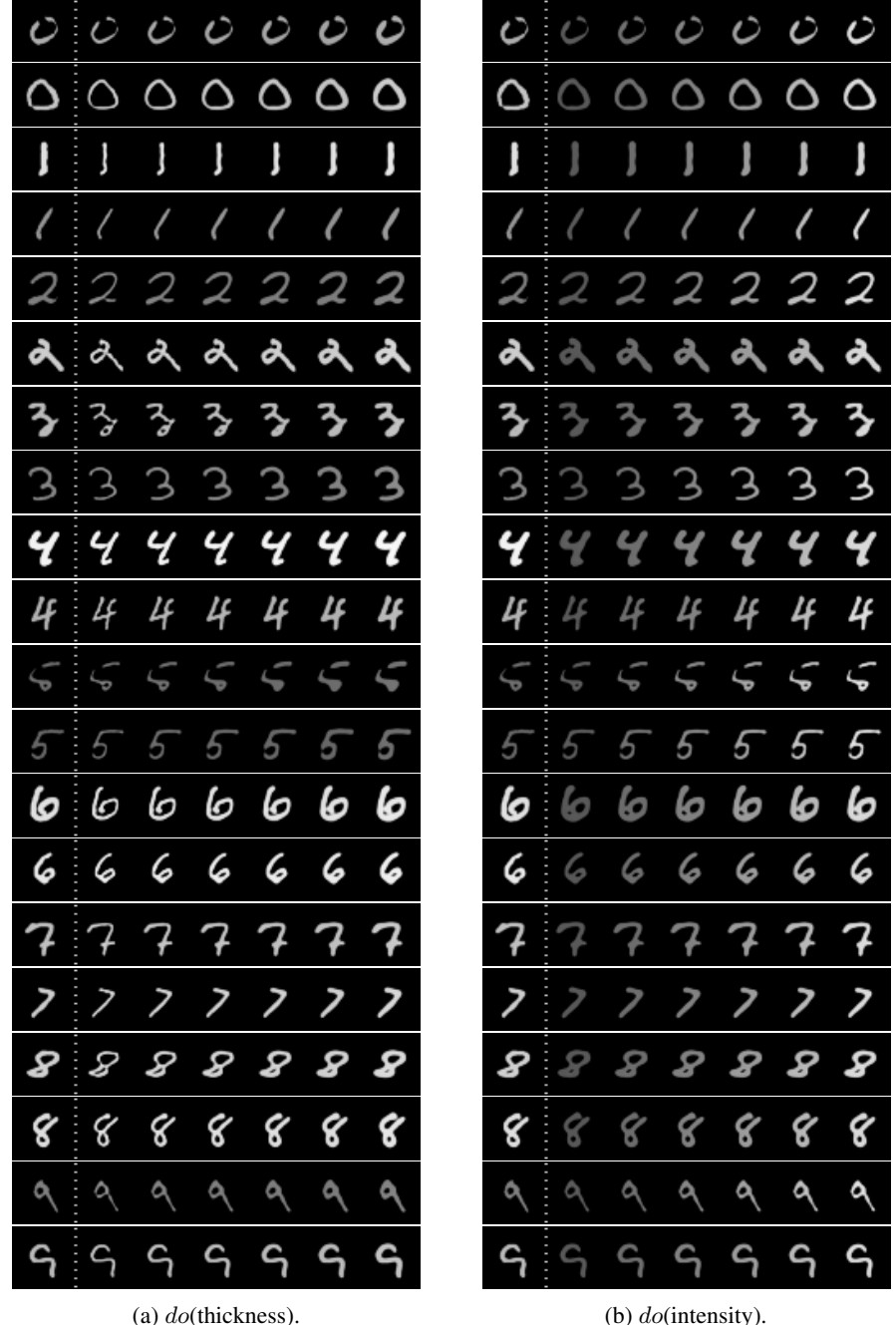

(a) *do*(thickness).  (b) *do*(intensity).

Figure 12: Sampled results on the test set of Morpho-MNIST. The left most image of each set is the original image. These results are non-cherry-picked in terms of model performance (picking was conducted to make sure there is a variety of digits and styles, but not based on performance).

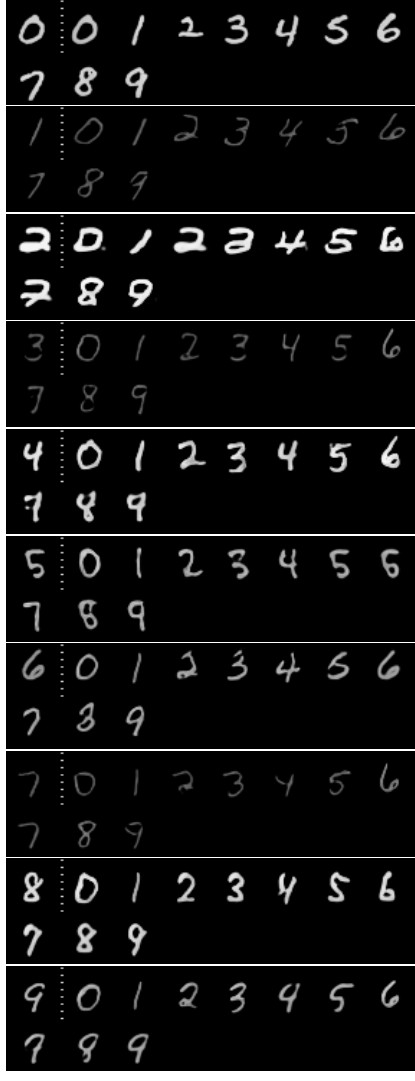

Figure 13: Sampled results on the test set of Morpho-MNIST on digit intervention. Note that these results do not have quantifiable counterfactual truths which we can evaluate on the capacity of intervening thickness and intensity in Table 2.

