# OpenReview forum: "Counterfactual Generative Modeling with Variational Causal Inference"
_ICLR.cc/2025/Conference — ICLR 2025 Poster_

### Official Review · Reviewer_nddM · 2024-10-30

**Soundness:** 3
**Presentation:** 3
**Contribution:** 3
**Rating:** 6
**Confidence:** 4

**Summary:**

This article presents a variational inference model specifically designed for counterfactual generation, motivated by the goal of extracting information contained within high-dimensional potential outcomes. This concept is intriguing, and the relevant experiments demonstrate the efficacy of the proposed method.

**Strengths:**

1.	The article tackles a vital issue in counterfactual generation, contributing meaningfully to the understanding of causal relationships in diverse fields.

2.	The focus on high-dimensional potential outcomes offers a reasonable angle, as it may capture complexities that traditional methods might miss, thus enhancing counterfactual generation.

3.	The experimental outcomes appear encouraging, indicating that the proposed model effectively supports counterfactual generation.

**Weaknesses:**

1.	The article does not provide asymptotic experiments to assess the performance of the proposed ELBO, which limits the understanding of its behavior as the sample size increases and may raise questions about its robustness in practical applications.

2.	The absence of a comparison between Morpho-MNIST and diffusion model-based counterfactual generation models Diff-SCM undermines the evaluation's robustness, making it difficult to assess the proposed model's effectiveness relative to state-of-the-art methods and diminishing its perceived impact within the field.

**Questions:**

1.	I don’t quite understand the sentence “hence preserving maximum individuality yet minimum treatment characteristics in counterfactual construction  g_theta(y, t')  during inference.” Could you explain it in more detail?

2.	L841. I have some questions regarding this statement. Why do you think that the performance of intervening with digits cannot be assessed? In fact, I find that evaluating the counterfactual performance of intervening with digits is more intuitive. I am also looking forward to seeing how your model performs with digit interventions.

3.	In your paper, you discuss the introduction of a novel ELBO. I am curious about its properties in terms of tightness. Specifically, how does this new lower bound compare to existing bounds in the literature? Additionally, what methods or analyses have you employed to assess its tightness, and how might this impact the overall performance and effectiveness of your model?

---

> ### Author Response · Authors · 2024-11-19
> **Response to Reviewer nddM**
>
> Thank the reviewer for the feedback and meaningful questions! We appreciate them and will do our best to address all your concerns. Per your request, we uploaded some results on intervening digits in Figure 13 (Appendix P.5). Note that intervening digits does not have quantifiable counterfactual truths which we can evaluate on the capacity of intervening thickness and intensity in Table 2. More details regarding this can be found in our answer to your Question 2 in the following paragraphs.
>
> ---
>
> Weakness 1 & Question 3:
>
> **Response**: We appreciate the reviewer’s unique perspective, however, we want to humbly and respectfully note that the reviewer might have a slight misunderstanding regarding variational lower bound (i.e. evidence lower bound, ELBO). The variational lower bound is very different from the bound of error or convergence that traditionally well-known in machine learning, it is simply a refactorization of an intractable objective using Bayes rule and Jensen’s inequality: same as any VAE-based variational lower bound, the inequality of VCI’s variational lower bound solely comes from Jensen's Inequality (see Eq. 29 on L1093 in the proof of Thm. 1), and we do not believe there is asymptotic analysis to perform here, and we have not have seen such analysis or experiment being performed for variational lower bound based methods. To the best of our knowledge, a tightness analysis is also rarely conducted in prior work that formulate around variational lower bound, including the original proposal of VAE. There were subsequent works of VAE we came across that discussed and improved on the tightness of variational lower bounds such as [1] and [2] by further factorizing the family of $q_\phi$ and $p_\theta$, but we respectfully think that this is quite out of the scope of this paper. The new lower bound is not a refinement on existing lower bounds, it is the same technique (Bayes rule and Jensen’s inequality) working towards a different objective (see Appendix B). However, we do not doubt that this tightness improvement could be a meaningful subsequent direction to pursue for VCI in future efforts.
>
> ---
>
> Weakness 2:
>
> **Response**: We appreciate this feedback because we think this is a crucial issue to address. It is very important to note that, *diffusion models are not state-of-the-art in counterfactual generative modeling*. It has not been shown that they have better capability in counterfactual generative modeling than HVAEs, because the diffusion mechanism fundamentally contradicts exogenous noise abduction, which is the key difference between conditional generative modeling and counterfactual generative modeling and the key difference between rung 2 and rung 3 causal inference on Pearl’s three layers of causality, as we discussed in Appendix B and particularly Appendix B.1. We think it is important to ask the question: “Are diffusion models automatically state-of-the-art in counterfactual generative modeling because of its capability in generative and conditional generative modeling?” Because counterfactual generative modeling is very different from conditional generative modeling, and as far as the former is concerned, MED [3] is definitively the state-of-the-art and diffusion models such as Diff-SCM has not shown nearly enough evidence on the level of MED in its capability of exogenous noise abduction (Appendix B.1.). This is to be expected because the diffusion process guarantees that exogenous noise is lost during encoding, so it is fundamentally ill-suited for counterfactual generative modeling, and one would actually be better off using a parameterized encoder as in a standard HVAE instead of the diffusion process. We very much understand it has become almost intuitive to think of diffusion models as state-of-the-art whenever image generations are concerned, hence why we put forward the lengthy discussion in Appendix B and Appendix B.1.
>
> We also want to highlight that, we specifically chose the state-of-the-art counterfactual generative modeling approach under the variational formulation (CHVAE, MED) and state-of-the-art under the adversarial formulation (DEAR) for benchmark comparison. We want to note that most prior works in this field have fairly weak baseline comparisons, commonly compared to one prior work in the field only, as is the case for Diff-SCM as well as the baselines we cited in Table 2. We made a significant effort to compare to state-of-the-arts in this field because we genuinely want to show that ours indeed outperforms the best CF generative models out there by a lot because it is theoretically well-motivated, and we tried to make sure the baseline selection reflects that. Therefore, we respectfully disagree with the statement that it “undermines the evaluation's robustness, making it difficult to assess the proposed model's effectiveness relative to state-of-the-art methods”. We respectfully believe we have adequately compared to state-of-the-art models in this field.

---

> ### Author Response · Authors · 2024-11-19
> **Answers to Questions**
>
> Question 1:
>
> **Response**: Thanks for the question and yes, we will elaborate in more detail. Suppose we supervise a generative model $g_\theta (y, t)$ only on observed outcome $y$, then the model is free to ignore the extra condition $t$ to converge to a state where $g_\theta (y, t) = g_\theta (y, 0)$, and if the latent dimension is large enough, it is free to simply converge to $g_\theta (y, t) = y$, meaning that it naively outputs the original image $y$ as counterfactual generation $g_\theta(y, t’)$ during inference. If this happens, the model “preserves maximum individuality” because all the individual details in the original image, i.e. exogenous noise, are perfectly kept in the counterfactual generation. However, it preserves “minimum treatment characteristics” because the counterfactual treatment $t’$ is totally not taken into consideration (take CelebA for example, this would be to return the original image of a non-smiling person when you inject the alternative condition “smiling”). We want to note that this might not be an issue for a generic autoencoder because the purpose is denoising and the bottleneck latent dimension is generally smaller than input and output dimensions, but it is particularly a problem for counterfactual generative modeling because denoising is *not* desired (ideally, we want to preserve all exogenous noise in the original input) and the latent dimensions in HVAEs and ours are at least as large as the input and output dimensions.
>
> ---
>
> Question 2:
>
> **Response**: Thanks again for the question! The results are shown in Figure 13 (Appendix P.5) of the newly uploaded manuscript. We want to humbly and respectfully point out that we did not quite say “the performance of intervening with digits cannot be assessed”, we said on L841-843 that “intervening digit does not have measurable counterfactual truth as intervening thickness and intensity in Morpho-MNIST” – in the Morpho-MNIST simulator, one can generate target counterfactual images with modified thickness and modified intensity, but cannot generate counterfactual images with modified digits. For example, the model decided to add a little imperfection to digit 3 and digit 8 in the counterfactual results of digit 6 based on the style of writing (see Figure 13), but there is no way to evaluate whether this would be true or not. This is to be expected because as we stated on L843-845, it is very hard and ambiguous to construct the counterfactual truth when one drastically changes the objects in the original images, and the Morpho-MNIST simulator certainly does not have such capability. In fact, when you drastically change the objects, the problem moves very far away from the consistency assumption (Assumption 2 on L1069), and it becomes really questionable whether the results under such “intervention” can even be defined as counterfactuals or not. If one only measures the digit classification accuracy of the counterfactual construction as a surrogate evaluation metric but not image MSE, as in MED [3], it would mean that they reduced the problem from rung 3 causal inference to rung 2 causal inference – it is no longer an evaluation of counterfactual generative modeling but a surrogate evaluation of interventional generative modeling or conditional generative modeling. We want to note that the Morph-MNIST dataset with thickness and intensity as interventions is the perfect simulation dataset for counterfactual generative modeling – it is a true interventional dataset where exogenous noise exists and counterfactual truths (for intervening thickness and intensity, not for intervening digits) can also be directly computed, through which we can quantitatively and simultaneously assess how much exogenous noise has been preserved and how much characteristics of the counterfactual intervention has been injected to the counterfactual construction. Intervening digits might be intuitive, but it reduces Morpho-MNIST to an observation dataset just like CelebA and one cannot quantitatively assess the preservation of exogenous.
>
> ---
>
> [1] Burda, Y., Grosse, R., and Salakhutdinov, R. Importance weighted autoencoders. In ICLR, 2016.
>
> [2] Rainforth, Tom, et al. "Tighter variational bounds are not necessarily better." In ICML, 2018.
>
> [3] Fabio De Sousa Ribeiro, Tian Xia, Miguel Monteiro, Nick Pawlowski, and Ben Glocker. High fidelity image counterfactuals with probabilistic causal models. 2023.

---

> > ### Comment · Reviewer_nddM · 2024-11-25
> >
> > Thank you for your detailed response. After reviewing it carefully, I have decided to retain my score.

---

### Official Review · Reviewer_81Hc · 2024-10-30

**Soundness:** 3
**Presentation:** 2
**Contribution:** 3
**Rating:** 6
**Confidence:** 5

**Summary:**

A framework is presented in which a counterfactual observation is estimated via a new variational formulation, in which the observed outcome and treatment are used within the conditional distribution for the unobserved (counterfactual) data. Experiments are presented on a range of applications, from biology to various types of images, with impressive quantitative performance.

**Strengths:**

The formulation is sound.  Figure 2 is particularly helpful in explaining the model and setup.

**Weaknesses:**

While the formulation is sound, the terminology of "causal reasoning" is questionable. One could argue that you are performing imputation of missing data, and then conditioning on the observed data makes complete sense. The criticism of the CEVAE is inappropriate, because they are solving a very different problem (traditional causal inference), and they do not condition on specific observations (the predictions are not "personalized"). It is ok to comment on and compare to CEVAE, but the setup and assumptions are different.

Another weakness, commented on below, is the presentation. The paper could be much better written, and thereby easier for the reader. Details below.

**Questions:**

1. In the beginning of Sec. 2.1, \Omega, \mathcal{Y}, \Sigma_Y, \mathcal{X}, \Sigma_{\mathcal{X}}, etc are not defined. This notation is unclear, and not clear why it is needed.

2. On p. 3, Assumptions 1 and 2, Remarks 1 and 2 are mentioned. The reader is very confused. None of these have been mentioned thus far, and the reader is not told such details are in the Appendix. This is unnecessarily confusing. By ICLR guidelines the main paper should be self contained. It is not.

3. On p. 3 it seems you use the adversarial discriminator D(X,T,Y) to manifest the loss. This seems ad hoc. The adversarial discriminator has a connection to the loss, but this is a surrogate and it is not particularly justified. I can see how it might work in practice, but this is an important point that deserves more discussion.

4. In (3), there is inconsistency in using upper case letters (Z for example) and lower case letters (e.g., z) for random variables. The way (2) useses notation on upper/lower case variables is not consistent with (3).

5. Does this really have to be framed as causal analysis? Why is it not just imputation of missing (counterfactual) data? I understand why you want to connect to methods like CEVAE, but they are doing something very different, and hence it seems unnecessary.

---

> ### Author Response · Authors · 2024-11-19
> **Response to Reviewer 81Hc**
>
> Thank the reviewer for the feedback! We appreciate your alternative perspective which provides us a valuable different angle of viewing counterfactual generative modeling. We will try to do our best to address all your concerns and answer all your questions in the following paragraphs.
>
> ---
>
> Weakness & Question 5:
>
> **Response**: Yes, having a rigorous causal formulation is absolutely necessary. It is the basis for formulating the causal objective and deriving the ELBO, and the basis for latent identifiability and disentanglement in this work. Without these bases, there will be no theoretically well-motivated disentangled exogenous noise abduction alongside counterfactual supervision, which is exactly what is missing in prior work. After all, our whole stochastic optimization scheme and all of our theoretical results are derived based on such causal formulation, so we do not think that this element is any redundant and we respectfully do not know how we are supposed to form the methodology in this work under a missing data imputation setup. We very much respect the reviewer’s alternative perspective, but there is evidently a certain level of rigorousness in causal formulation that is standard and to be expected in prior work in the counterfactual generative modeling community [1][2][3]. Counterfactual inference *is* rung 3 causal inference, and we respectfully do not feel that there is causal terminology used in this work that is questionable. If this is not the case and there is any terminology in specific that the reviewer finds inappropriate or unsound, we humbly ask the reviewer to point it out and we will be happy to correct any mistake. Regarding CEVAE (as well as CausalGAN), we do not particularly want to relate our work to it, as you are right about its difference in purpose (although CEVAE is in fact working towards *individualized treatment effect*, i.e. “personalized” to the covariate level). As a matter of fact, we would personally prefer leaving CEVAE out of the comparison entirely and focus on comparison to DEAR, CHVAE and MED, but we made the judgment that this is not possible because for readers with slightly less background in causal machine learning than the reviewer, comparison to CEVAE is one of the first thing they look for (e.g. see reviewer m2tW’s comment) and from our experience it is quite impossible to leave CEVAE out of discussion even if it is not particularly relevant. Besides, we believe such comparison also contributes to a more thorough discussion of prior work. Therefore, we respectfully hope the reviewer understands the decision to leave CEVAE in the comparisons.
>
> *“the criticism of the CEVAE is inappropriate”*
>
> **Response**: We apologize that it came across this way but after re-examining the paper, we failed to find a comment in our manuscript that criticizes CEVAE. If there is any specific content the reviewer would like us to change regarding CEVAE, could you please kindly let us know? We will make the adjustment. We agree with the reviewer that “it is ok to comment on and compare to CEVAE” – in the manuscript, we simply left CEVAE in the comparisons without excessive comments. Per the reviewer’s suggestion, we have added a sentence on L220 to clarify its difference in setup.
>
> ---
>
> [1] Nick Pawlowski, Daniel Coelho de Castro, and Ben Glocker. Deep structural causal models for tractable counterfactual inference. NeurIPS, 2020.
>
> [2] Pedro Sanchez and Sotirios A Tsaftaris. Diffusion causal models for counterfactual estimation. CLeaR, 2022.
>
> [3] Fabio De Sousa Ribeiro, Tian Xia, Miguel Monteiro, Nick Pawlowski, and Ben Glocker. High fidelity image counterfactuals with probabilistic causal models. ICML, 2023.

---

> ### Author Response · Authors · 2024-11-19
> **Answers to Questions**
>
> Question 1:
>
> **Response**: We have removed all these notations per the reviewer’s suggestion. Please kindly check out the updated manuscript.
>
> ---
>
> Question 2:
>
> **Response**: We have added a sentence on L131-132 to more clearly indicate that detailed assumptions necessary for the definition of counterfactual can be found in the Appendix before referring to them in order to reduce confusion, as you suggested. Thank the reviewer for raising this issue.
>
> ---
>
> Question 3:
>
> **Response**: We respectfully believe that this is justified. The root purpose of adversarial training is distribution alignment and this is exactly what term 2 of the variation lower bound (Eq. 2) suggests: $D \left[ p (Y | X, T) \parallel p (Y' | X, T') \right]$, where the counterfactual outcome log-likelihood estimated by the discriminator corresponds to $\log \left[ \hat{p} (y_{\theta,\phi}' | x, t') \right]$. The estimation of term 2 ($\log \left[ \hat{p} (y_{\theta,\phi}' | x, t') \right]$) is surrogate either way: even if one pre-trains a neural $\hat{p}$ model not in an end-to-end fashion, the counterfactual supervision is still a surrogate model estimate of the interventional log-likelihood.
>
> ---
>
> Question 4:
>
> **Response**: Thanks for the feedback! This notation is intended and consistent: the upper case letter indicates the random variable and the lower case letter indicates a value/realization in the codomain of the random variable. The estimating models do not directly take random variables as inputs. This notation is quite standard in prior works formulated under Pearl’s NPSEM formulation [4] as well as the line of work in TMLE [5] that we referred to for deriving the efficient influence function. We believe that this reduces confusion compared to using all upper/lower case notations, and a notable example of how using all upper/lower cases would have drastically increased confusion can be found in the proof of Theorem 2 (page 26).
>
> ---
>
> [4] Judea Pearl. Causal diagrams for empirical research. Biometrika, 82(4):669–688, 1995.
>
> [5] Mark J Van Der Laan and Daniel Rubin. Targeted maximum likelihood learning. The international journal of biostatistics, 2(1), 2006.

---

### Official Review · Reviewer_QZf3 · 2024-11-03

**Soundness:** 3
**Presentation:** 3
**Contribution:** 3
**Rating:** 6
**Confidence:** 4

**Summary:**

This work studies counterfactual generation through a newly proposed variational inference paradigm specific to causal models. The proposed Variational Causal Inference (VCI) framework considers the generation of counterfactual outcomes relying on the consistency assumption, where potential outcomes and the true outcomes are the same under a counterfactual treatment. The authors propose a data-generating process from a counterfactual perspective where the latent variables derived from covariates, treatment, and outcome are aligned with latent variables derived from counterfactual outcomes. Utilizing the new counterfactual formulation, an appropriate variational lower bound is derived. The authors show theoretical identifiability guarantees of the proposed formulation. Empirical results show the effectiveness of the VCI framework in generating accurate counterfactuals.

**Strengths:**

- Overall, I believe this work sets forth a very interesting formulation for counterfactual inference in generative models and criticizes the use of conditional generative models for counterfactual generation.
- The VCI framework is formulated well and the consistency assumption to ensure counterfactuals keep individual components of the factual but also reflect changes according to a treatment is intuitive.
- The theoretical results for ELBO derivation, identifiability, and disentanglement further strengthen the paper and properly contextualize the work within the broader literature of causal generative models.
- The empirical results show the capability of the proposed framework to generate accurate counterfactuals on several datasets including image and single-cell perturbation datasets.

**Weaknesses:**

- It seems that the Effectiveness metric from [1] is utilized through the MAE, where an anti-causal predictor is trained to predict the values of thickness and intensity, and a counterfactual reconstruction is fed in for evaluation. I believe this to be a sound metric. However, I do not think the MSE between the true and generated images is necessarily a good metric to evaluate the quality of the counterfactual. The MSE evaluates a pixel-wise similarity between the two images. However, in a counterfactual, the changes are more in the abstract variables describing the image.
- Although there are theoretical intuitions for disentanglement, there is no strict evaluation of the latent disentanglement of z. The DCI metric [2] evaluates the level of one-to-one correspondence between latent factors and the ground truth generative factors. I am not sure if this metric applies in evaluating disentanglement between z and the treatment variable t, but it would certainly be good to use a similar metric (perhaps Mean Correlation Coefficient) to evaluate disentanglement.

[1] Monteiro et al. Measuring axiomatic soundness of counterfactual image models. ICLR 2023.

[2] Eastwood and Williams. A Framework for the Quantitative Evaluation of Disentangled Representations. ICLR 2018.

**Questions:**

- There does not seem to be any explicit modeling of causal mechanisms in the proposed framework. For example, the causal graph thickness → intensity can be imposed on the MorphoMNIST dataset. Previous work, such as DeepSCM [3], CausalHVAE [4], and CausalDiffAE [5], explicitly model the causal mechanism between the two variables. For systems with complex causal relations, can the proposed framework model the causal mechanisms?
- The notion of disentanglement in this work is quite different than in disentangled representation learning from non-linear ICA based approaches. If T is observed, what is the intuitive meaning behind disentangling the latent factors Z from T? It is not clear to me why enforcing the latent space loss yields a disentangled objective.

[3] Pawlowski et al. Deep Structural Causal Models for Tractable Counterfactual Inference. NeurIPS 2020.

[4] Ribeiro et al. High Fidelity Image Counterfactuals with Probabilistic Causal Models. ICML 2023.

[5] Komanduri et al. Causal Diffusion Autoencoders: Toward Counterfactual Generation via Diffusion Probabilistic Models. ECAI 2024.

---

> ### Author Response · Authors · 2024-11-19
> **Response to Reviewer QZf3**
>
> We want to thank the reviewer for the constructive feedback. From your feedback, we can see that the reviewer has thought through their questions before asking them, and it helps make the discussion more collaborative and effective, which we deeply appreciate. We will strive to thoroughly answer all your questions and try our best to address any concern you have in our rebuttal.
>
> ---
>
> Weakness 1:
>
> **Response**: We would like to maintain the opinion that image MSE is the best metric to evaluate counterfactual generations when counterfactual truths are attainable (not perfect, but best among available metrics), and we would like to explain to the reviewer our reasoning. To make the explanation short: factors’ MAE (it’s not exactly effectiveness in [1]: effectiveness is essentially a surrogate estimate of factors’ MAE when counterfactual truths are not attainable) evaluates interventional inference (rung 2 causal inference), while image MSE evaluates counterfactual inference (rung 3). Factors’ MAE or effectiveness only measures how much characteristics of the *observed factor* is shown in the counterfactual construction, it does not take into account how much exogenous noise is preserved in the construction, which is the definitive difference between interventional inference and counterfactual inference. To give the reviewer an example, if one only evaluates observed factors’ MAE as in MED ([2], they used digits as observed factors as well and surrogate classification accuracy for effectiveness), then the best model would be the model that simply produces the average and generic outcome under observed factors: say one would like to know what a digit 8 under 1.2 thickness and 3.4 intensity would have looked like if the writer wrote with 5.6 intensity, the best model would be a model that simply gives you an average image of digit 8 with 1.2 thickness and 5.6 intensity. Any further details (i.e. exogenous noise) that one would like to preserve – the style of the writing, the unique curves, the accidental gaps, etc. would not matter as the evaluation metrics completely ignore them. But obviously, this model is meaningless when the scope is counterfactual inference rather than interventional inference. We would like to reiterate that we agree with the reviewer that image MSE is not perfect, but it is simply the best metric available when counterfactual truths are attainable to the best of our knowledge, and the only metric that truly evaluates rung 3, counterfactual inference.
>
> ---
>
> Weakness 2:
>
> **Response**: We have added such evaluation results in Appendix P.3 in the newly uploaded manuscript per the reviewer’s suggestion. You are right, the DCI metric is not particularly applicable here, because $z$ is only meant to be disentangled with respect to the treatment factor $t$ rather than mapped to generative factors. We think the best metric to use in this context is to evaluate by definition of disentanglement under our context (definition 1 on L982, extended from [3]), i.e. the cycle-consistency (i.e. oracle consistency) of $z$ when $t$ is intervened, and we presented these results in Table 6 of Appendix P.3.
>
> [1] Miguel Monteiro, Fabio De Sousa Ribeiro, Nick Pawlowski, Daniel C Castro, and Ben Glocker. Measuring axiomatic soundness of counterfactual image models. 2023.
>
> [2] Fabio De Sousa Ribeiro, Tian Xia, Miguel Monteiro, Nick Pawlowski, and Ben Glocker. High fidelity image counterfactuals with probabilistic causal models. 2023.

---

> ### Author Response · Authors · 2024-11-19
> **Answers to Questions**
>
> Question 1:
>
> **Response**: Yes. As stated on L535-539 in the conclusion section, explicit modeling of causal relations between observed factors can be trivially integrated into our framework – one simply needs to update the observed factors under intervention before feeding them into our outcome construction framework. To put in other words, the proposed method can be naively integrated with DeepSCM – one just needs to replace the outcome construction model in DeepSCM with ours, and all the modeling of observed factors in DeepSCM remains the same. Although it would be a good direction to pursue for practical reasons, we think that it is out of the scope of the key contribution we would like to deliver and we would prefer to focus on presenting the outcome construction framework itself that we proposed in this work. Also note that Morpho-MNIST is a true interventional dataset, and the ability to implicitly model thickness -> intensity relation is captured by the evaluation metrics (col. 3 and col. 9 of Table 2). So if the method does a poor job of implicitly modeling such relations, it will be reflected on these metrics.
>
> ---
>
> Question 2:
>
> **Response**: The notion of disentanglement is defined as the cycle-consistency or oracle-consistency of $z$ and $t$ when the counterpart is modified (definition 1 on L982) – this is a well-accepted notion of disentanglement first defined in [3], and you are right, it is quite different from the notion of disentanglement in non-linear ICA, as we stated in Appendix A. To put this definition into intuitive words: for a latent $z$ and any two different treatment $t$ and $t’$, if the encoder $q_\phi$ is able to recover the same latent for the constructed outcome under $(z, t)$ and constructed counterfactual outcome under $(z, t’)$, then $z$ is oracle-consistent – this is what Lemma 1 is and why encouraging the encoder to recover the same latent distribution under factual and counterfactual outcomes (i.e. the latent space loss) yields latent disentanglement by definition. To give the reviewer another intuitive example under CelebA: for two images of the same person, one with glasses ($t$) and one without glasses ($t’$), if the encoder recover the same latent distribution for these two different images, that means the latent does not encode information about glasses, i.e. $z$ and $t$ are disentangled. In light of your question, we have changed the terminology from “latent disentanglement” to “disentangled abduction of exogenous noise” in various places in our paper to make the terminology more specific and hopefully cause less confusion.
>
> [3] Rui Shu, Yining Chen, Abhishek Kumar, Stefano Ermon, and Ben Poole. Weakly supervised
> disentanglement with guarantees. 2019.

---

> > ### Comment · Reviewer_QZf3 · 2024-11-25
> >
> > I appreciate the authors taking the time to provide clarifications and answer my questions. I think cycle-consistency disentanglement with respect to exogenous noise abduction rather than latent disentanglement is helpful and provides more clarity. Although I believe that the evaluation is a bit weak and the metric evaluations using MSE are not necessarily a good indicator of more accurately generated counterfactuals, I think the framing is quite neat and shows a fundamental variational approach to developing counterfactual generative models. Therefore, I maintain my score leaning toward acceptance of this paper.

---

> ### Author Response · Authors · 2024-11-25
> **Response to Reviewer QZf3**
>
> Thank the reviewer for the appreciation of our clarification. If the reviewer has a better metric in mind to evaluate counterfactual generation and exogenous noise abduction, could you kindly let us know? We humbly and respectfully think that criticizing the MSE metric without an alternative does not leave us a viable action. As we discussed, we totally agree with the reviewer that MSE is not perfect, but in this setting where counterfactual truths are attainable and unobserved exogenous factors are present, there is simply no better metric than MSE to evaluate rung 3, counterfactual inference. We also presented the MAE on the observed abstract factors along with MSE, and we respectfully do not quite know what else is better to present here.
>
> Besides, if we strictly stick to the formal definition of counterfactual — it should share the exact same copy of exogenous as the observed image — that means every single pixel that is not causally affected by the treatment intervention should stay exactly the same. By that standard, evaluation by image MSE is very much aligned with the purpose of counterfactual inference.

---

> > ### Comment · Reviewer_QZf3 · 2024-11-25
> >
> > I thank the authors for the response. To clarify, I have merely pointed out that the MSE may not be a good metric for L3 evaluation. However, I do recognize that there is a lack of metrics for evaluating such a quantity, especially for data such as images. Thus, I do not fault the authors for using MSE for evaluation. My main concern is that the bulk of the experiments are conducted on the MorphoMNIST dataset. I acknowledge that we do not have counterfactual truth for CelebA, but it would be more convincing to use surrogate methods such as composition, reversibility, and effectiveness for evaluation. There have been other works that benchmark counterfactual generation using these metrics [1].
> >
> > [1] Melistas et al. Benchmarking Counterfactual Image Generation. NeurIPS 2024.

---

> ### Author Response · Authors · 2024-11-28
> **Response to Reviewer QZf3**
>
> Thank the reviewer for the response. We think your clarification here is very sensible and reasonable, and thus we have run composition, reversibility and effectiveness metrics on our models as well as the best benchmark models in Table 2. The results are shown in Appendix P.4 of the updated manuscript. We hope that this addition could help the reviewer reach the score of a firm acceptance. We want to respectfully note that, although we find the reviewer’s response to be very sensible, this clarification was posted on 11/25, which unfortunately did not leave us enough time to tune and run benchmark models on CelebAHQ (it is much slower than MorphoMNIST) and measure the three metrics in order to present the results before the 11/27 paper revision deadline. From the initial review (Weakness 1), it wasn’t clear to us that what the reviewer wants is results on a different dataset rather than results on a different metric. As the email that was sent out earlier on 11/25 by PCs suggested, “*Reviewers are instructed to not ask for significant experiments, and area chairs are instructed to discard significant reviewer requests.*”. We want to emphasize that this is not by any mean a knock against the reviewer – we think the reviewer has given us constructive feedback and clearly has great expertise in causal machine learning from these responses – we simply want to highlight that it would be slightly unfair to expect such results on CelebAHQ given the short notice.
>
> As for the results on composition, reversibility and effectiveness in Appendix P.4, we want to note that these are great metrics for model diagnosis purposes but inconclusive for the purpose of deciding the best counterfactual generative model (also note that composition and reversibility are both pixel-level MSE quantities just like image MSE). For example, as we discussed in paragraph 2 of Appendix M, a model that *does nothing* and simply returns the original input image as counterfactual outcome would achieve the best score on 2 of these 3 metrics (this is not a speculation, it is analytically true) – a composition of 0 and a reversibility of 0. Therefore, unless there is some sort of utility function, or one model dominates the other model on all 3 metrics, it would be hardly conclusive for anyone to tell which is the best counterfactual generative model by looking at these three columns.
>
> For example, the composition metric does not interact with the counterfactual intervention label at all (while at least the other two metrics interact with intervention on some capacity), and a composition of 0 merely means the model can do exact reconstruction of the original image – it does not reflect model’s counterfactual generative ability unless weighed with other factors. As discussed on L479-485, it is expected that our model does not reach a composition of 0 because it does not necessarily perform exact reconstruction on the regions directly affected by the intervention – which is not a problem for the purpose of counterfactual construction. Therefore, for the purpose of deciding the best counterfactual generative model, we firmly believe that conducting quantitative evaluations on a *true interventional dataset* where counterfactual truths are attainable such as MorphoMNIST is the best option in comparison to surrogate quantitative results on observational datasets such as CelebAHQ. And from results shown in Table 2, Table 8, Figure 12 and Figure 13, we believe that there is strong evidence that our model has *vastly* outperformed state-of-the-arts.

---

> > ### Comment · Reviewer_QZf3 · 2024-11-28
> >
> > I thank the authors for their efforts to include additional clarifications and results. I wanted to see the evaluations from composition, reversibility, and effectiveness as I do not believe it would take much time to run these evaluations on the datasets and I think they are important metrics to report. However, I acknowledge that the rebuttal period is short and I appreciate the authors taking the time to run these evaluations for MorphoMNIST. If possible, I would suggest including more quantitative evaluations for the CelebA-HQ dataset in a later draft. I will update my score after discussion with other reviewers. I am currently leaning toward accepting this paper.

---

> ### Author Response · Authors · 2024-11-28
> **Response to Reviewer QZf3**
>
> Yes, we can certainly do that in the final revision as we do think that this is reasonable. Thank the reviewer for the understanding!

---

### Official Review · Reviewer_m2tW · 2024-11-06

**Soundness:** 2
**Presentation:** 1
**Contribution:** 1
**Rating:** 3
**Confidence:** 4

**Summary:**

This paper studies counterfactual generative modeling in terms of neural variational causal inference. The authors claimed that they proposed a new VAE formulation for this purpose with fundamental changes and achieved latent disentanglement through distribution alignment. Several experiments are conducted to demonstrate the author's claim.

**Strengths:**

The work is theoretically grounded.

**Weaknesses:**

The manuscript should properly discuss the previous works, such as CEVAE, in the main paper.

Figure 1 could be more specified with additional links that indicate the inference network in the VAEs.

If the proposed method has novelty or contribution in terms of latent identifiability and disentanglement, appropriate experiments should be conducted with fair comparison against the works within such sub-fields.

What I am concerned about is the following. I found that this paper is an extension of a workshop paper where some theoretical results and experimental results are almost duplicated from the previous version. While the recent AI community is rapidly growing and being generous in such situations, I don't think this is the right direction for the development of academia.

**Questions:**

None

---

> ### Author Response · Authors · 2024-11-19
> **Response to Reviewer m2tW**
>
> Thank the reviewer for the feedback! First and foremost, we would like to address your concern regarding NeurIPS workshop paper “Variational Causal Inference”. We want to kindly note to the reviewer that NeurIPS workshop paper does not count as formal publication, and continuing works from NeurIPS workshop are allowed to be submitted to ICLR main conference (https://iclr.cc/Conferences/2025/FAQ). After all, the purpose of NeurIPS workshop is to provide an informal venue for in-person discussion of *work in progress* (https://neurips.cc/Conferences/2024/CallForWorkshops). Without revealing any identity information, we would like to suggest to the reviewer that the authors might have always planned to submit the in-progress manuscript to a conference workshop to first gather the community’s feedback – which is exactly what NeurIPS workshops are set to do – before finishing it and submitting to a main conference, and would be very confused that they are getting punished because of it. Important advancements to “Variational Causal Inference” have been made in this work and we have summarized them in Appendix M. We want to kindly and respectfully note to the reviewer that we do not expect “generosity from the AI community” for the evaluation of this work, as this practice indeed follows the guidelines to the best of our knowledge and does not lie in some gray area. If this is not the case, we are happy to get desk-rejected by the ACs. That being said, we humbly and sincerely hope the reviewer could reconsider the score and give us an evaluation based on the quality of this work and the proposed method.
>
> ---
>
> Weakness 1: *The manuscript should properly discuss the previous works, such as CEVAE, in the main paper.*
>
> **Response**: Related work including CEVAE have been discussed and thoroughly compared to in Appendix C, and the direction to the related work section was provided to the readers at the end of the introduction (L97). In addition, we have highlighted L97 in the newly uploaded manuscript to make it stand out. Given that the proposed method sets forth a completely different direction compared to previous work in counterfactual generative modeling (this is also noted by Reviewer QZf3) and the page limitation, we have made the judgment to put this section in the Appendix to best allocate the content of presentation.
>
> ---
>
> Weakness 2: *Figure 1 could be more specified with additional links that indicate the inference network in the VAEs.*
>
> **Response**: Thanks for the feedback. We want to respectfully note to the reviewer that Figure 1 is the problem formulation, or causal formulation, and it is not related to model workflow, neural architecture or inference network. The notion of inference network does not exist in VCI, both the encoder $q_\phi$ and decoder $p_\theta$ are used during inference time and the counterfactual construction workflow is not different from training time (in fact, this is one of the key innovations: to construct counterfactuals during training time just like the way you would during inference time and conduct counterfactual supervision on these constructions. To the best of our knowledge, no prior work constructs and supervises counterfactuals end-to-end during training time). We would like to direct the reviewer to Figure 2 for model workflow in stochastic optimization, and Figure 8 if you would like to see the model architecture for image generation tasks. As commented by Reviewer 81Hc, Figure 2 is particularly helpful in explaining the model and setup.
>
> ---
>
> Weakness 3: *If the proposed method has novelty or contribution in terms of latent identifiability and disentanglement, appropriate experiments should be conducted with fair comparison against the works within such sub-fields.*
>
> **Response**: To the best of our knowledge, there isn’t any prior work in this field that conducts disentangled abduction of exogenous noise. The term “latent disentanglement” is a broad summarization, and as pointed out by Reviewer QZf3, the latent disentanglement conducted by works in non-linear ICA (which is what researchers have primarily focused on in prior works in causal representation learning, see Appendix A) is very different from the latent disentanglement performed in ours, as they serve completely different purposes and are not comparable, despite the terminology might have made them sound similar. We have changed the terminology in this paper from describing the latent divergence term as “latent disentanglement” to a more specific terminology “disentangled abduction of exogenous noise”, in order to eliminate such confusions. To summarize, we did not compare our disentangled abduction of exogenous noise to prior works in this sub-field because there isn’t any prior work in this sub-field to compare to. In such cases where there is no prior work to compare to, we think it is incredibly important to compare it in ablation studies, which is what we did in Figure 3, Figure 4 and Table 6.

---

### Meta-Review · Area_Chair_TfMU · 2024-12-20

**Metareview:**

The authors look into counterfactual generation through by developing a variational lower bound on the counterfactual log probability in a framework called variational causal inference. Identifiability was discussed in connection with noise disentanglement.  The experiments across multiple domains were positive. The reviewers were all positive and confident (except for one reviewer whose main concern was about a dual submission).

**Additional Comments On Reviewer Discussion:**

Three reviewers were positive. The one negative reviewer mentions a similar workshop submission, but this work does not appear to have been published at a full conference.

---

### Decision · Program_Chairs · 2025-01-22

Accept (Poster)